

# Modelling Atmospheric Mineral Aerosol Chemistry to Predict Heterogeneous Photooxidation of SO$_2$

Zechen Yu, Myoseon Jang, and Jiyeon Park

P.O. Box116450, Department of Environmental Engineering Sciences, Engineering School of Sustainable
Infrastructure and Environment, University of Florida, Gainesville, FL, USA, 32611

Corresponding author: Myoseon Jang, mjang@ufl.edu

**Abstract.**

The photocatalytic ability of airborne mineral dust particles is known to heterogeneously promote SO$_2$ oxidation, but prediction of this phenomenon is not fully taken into account by current models. In this study, the Atmospheric Mineral Aerosol Reaction (AMAR) model was developed to capture the influence of air–suspended mineral dust particles on sulfate formation in various environments. In the model, SO$_2$ oxidation proceeds in three phases including the gas phase, the inorganic–salted aqueous phase (non–dust phase), and the dust phase. Dust chemistry is described as the adsorption–desorption kinetics (gas–particle partitioning) of SO$_2$ and NO$_x$. The reaction of adsorbed SO$_2$ on dust particles occurs *via* two major paths: autoxidation of SO$_2$ in open air and photocatalytic mechanisms under UV light. The kinetic mechanism of autoxidation was first leveraged using controlled indoor chamber data in the presence of Arizona Test Dust (ATD) particles without UV light, and then extended to photochemistry. With UV light, SO$_2$ photooxidation was promoted by surface oxidants (OH radicals) that are generated *via* the photocatalysis of semiconducting metal oxides (electron–hole theory) of ATD particles. This photocatalytic rate constant was derived from the integration of the combinational product of the dust absorbance spectrum and wave–dependent actinic flux for the full range of wavelengths of the light source. The predicted concentrations of sulfate and nitrate using the AMAR model agreed well with outdoor chamber data that were produced under natural sunlight. For seven consecutive hours of photooxidation of SO$_2$ in an outdoor chamber, dust chemistry at the low NO$_x$ level was attributed to 70% of total sulfate (60 ppb SO$_2$, 290 μg m$^{-3}$ ATD, and NO$_x$ less than 5 ppb). At high NO$_x$ (>50 ppb of NO$_x$ with low hydrocarbons), sulfate formation was also greatly promoted by dust chemistry, but it was significantly suppressed by the competition between NO$_2$ and SO$_2$ that both consume the dust–surface oxidants (OH radicals or ozone). The AMAR model, derived in this study with ATD particles, will provide a platform for predicting sulfate formation in the presence of authentic dust particles (e.g. Gobi and Saharan dust).



## 1 Introduction

The surface of mineral dust particles is able to act as sink for various atmospheric trace gases such as sulfur dioxide ($SO_2$), nitrogen oxides ($NO_x$, e.g. NO and $NO_2$), and ozone ($O_3$). Among trace gases, $SO_2$ has received much attention because heterogeneous oxidation of $SO_2$ produces nonvolatile sulfuric acid, which is readily involved in the acidification of particles or the reaction with dust constituents such as alkaline metals ($K^+$, $Na^+$) or metal oxides (e.g. $\alpha$–$Al_2O_3$ and $Fe_2O_3$). Such modification of the chemical composition of dust particles can influence the hygroscopic properties of mineral dust, which is essential to activate cloud condensation nucleation (Krueger et al., 2003; Zhang and Chan, 2002; Vlasenko et al., 2006; Liu et al., 2008; Tang et al., 2016).

Metal oxides (e.g. $TiO_2$ and $Al_2O_3$) have frequently been used in many laboratories to study the key role of mineral dust in the heterogeneous oxidation of $SO_2$ (Goodman et al., 2001; Usher et al., 2002; Zhang et al., 2006). However, these laboratory studies have been limited to a certain type of metal oxide and autoxidation of $SO_2$ without a light source. To date, only a few studies have attempted to study the photocatalytic characteristics of mineral dust in the oxidation of $SO_2$ and $NO_x$. For example, as noted by Park and Jang (2016), the reactive uptake coefficient ($\gamma_{SO_4^{2-}}$) of $SO_2$ in the presence of dry Arizona Test Dust (ATD) particles under UV light was one order of magnitude higher ($1.16 \times 10^{-6}$ using an indoor chamber with a light mix of UV–A and UV–B light) than that from autoxidation ($1.15 \times 10^{-7}$) without a light source. Using an aerosol flow tube, Dupart et al. (2014) observed that the uptake rate of $NO_2$ by ATD dust particles was significantly enhanced (by four times) under UV–A irradiation compared to dark conditions. Field observations have also reported the promotion of $SO_2$ photooxidation in the presence of mineral dust. For instance, near Beijing, China (ground–based campaign in 2009), and in Lyon, France (remote–sensing campaign in 2010), Dupart et al. (2012) found that mineral dust was a source of OH radicals under UV radiation that promoted sulfate formation.

Semiconducting metal oxides (e.g. $\alpha$–$Al_2O_3$, $\alpha$–$Fe_2O_3$, and $TiO_2$) act as a photocatalyst in mineral dust particles that can yield electron ($e^-_{cb}$)–hole ($h^+_{vb}$) pairs, and that they are involved in the production of strong oxidizers, such as superoxide radical anions ($O_2^-$) and OH radicals (Linsebigler et al., 1995; Hoffmann et al., 1995; Thompson and Yates, 2006; Cwiertny et al., 2008; Chen et al., 2012; Dupart et al., 2014; Colmenares and Luque, 2014). These oxidizers enable rapid





oxidation of adsorbed $SO_2$ and $NO_x$ on the surface of mineral dust particles. For example, using transmission FTIR spectroscopy and X–ray photoelectron spectroscopy, Nanayakkara et al. (2012) observed the oxidation of $SO_2$ by the photo–catalytically generated OH radicals in the presence of titanium oxide particles. The heterogeneous formation of sulfate and nitrate can be highly variable

and dependent on the chemical characteristics of dust aerosol (Gankanda et al., 2016). Authentic mineral dust particles differ from pure metal oxides in chemical composition. For example, Wagner et al. (2012) reported that the content of metal oxides in Saharan dust samples from Burkina Faso includes 14% $Al_2O_3$, 8.4% $Fe_2O_3$, and 1.2% $TiO_2$.

       Most research on dust photochemistry has been limited to qualitative studies and lacks

kinetic mechanisms that are linked to a predictive model. The typical wave–dependent photolysis of gas–phase trace gases has long been subject to atmospheric photochemistry. This photolysis rate is a first–order reaction and is calculated *via* the coupling actinic flux (the quantity of photons) with the characteristics (cross section area and quantum yield) of a light–absorbing molecule (McNaught and Wilkinson, 1997). In order to model dust photochemistry, the integration of

wavelength–dependent actinic flux with photocatalytic activity of mineral dust is needed.

       In addition to sunlight intensity, humidity also influences heterogeneous dust chemistry. Humidity governs particle water content, which influences the gas–dust sorption process of trace gases (Navea et al., 2010) and the formation of dust–phase oxidants. Huang et al. (2015) found that the $\gamma_{SO_4^{2-}}$ of $SO_2$ autoxidation in ATD particles increased by 142% because of the relative

humidity (RH) changed from 15% to 90%. In the presence of UV light, the particle water content can act as an acceptor for $h^+_{vb}$ and produce surface OH radicals, promoting heterogeneous photochemistry of $SO_2$ on mineral dust. In the presence of UV light, Shang et al. (2010) reported that sulfate production on the surface of $TiO_2$ increased by five times because of the increase of RH from 20% to 80%. Park and Jang (2016) also reported the exponential increase in $\gamma_{SO_4^{2-}}$ as the

RH increased from 20% to 80% for both autoxidation and photooxidation of $SO_2$ in the presence of ATD particles. A few studies have attempted to simulate sulfate formation in the presence of mineral dust at regional scales using laboratory–generated kinetic parameters (Tang et al., 2004; Li and Han, 2010; Dong et al., 2016). However, $\gamma_{SO_4^{2-}}$ applied to the regional simulations originated from pure and dry metal oxides without UV light, and thus will differ from those of

ambient dust exposed to natural sunlight. It is expected that the typical regional simulations during dust events might underestimate the formation of sulfate.





In this study, the Atmospheric Mineral Aerosol Reaction (AMAR) model was developed to predict atmospheric oxidation of trace gases such as $SO_2$ and $NO_2$ under ambient conditions. The kinetic mechanisms of dust–driven photochemistry, including autoxidation and photooxidation of $SO_2$, was newly established in the model. The rate constant of dust

photoactivation, which forms electron–hole pairs and sources dust–driven oxidants, was integrated into the model. The influence of meteorological variables, such as humidity, temperature and sunlight, on $SO_2$ oxidation was investigated using the resulting AMAR model. The model also addresses the kinetic mechanism to simulate how atmospheric major pollutants such as $NO_x$ and ozone are engaged in the oxidation of $SO_2$ in the presence of airborne dust particles. For

environmental scenarios, the model was applied for polluted urban conditions (e.g. hydrocarbon ppbC/$NO_x$ ppb < 5) and low $NO_x$ conditions (e.g. hydrocarbon ppbC/$NO_x$ ppb > 5). The reaction rate constants for both autoxidation and photocatalytic reactions of $SO_2$ were obtained through the simulation of indoor chamber data, which were previously generated under various meteorological and environmental conditions (Park and Jang, 2016). The suitability of the resulting AMAR model

was tested against sulfate formation in a large outdoor smog chamber at the University of Florida Atmospheric Photochemical Outdoor Reactor (UF–APHOR) under natural sunlight. The AMAR model of this study will vastly improve the accuracy of the prediction of sulfate and nitrate formation in regional and global scales where dust emission is influential.

## 2 Experimental section

**2.1 Chamber experiments**

The indoor chamber data of this study was obtained from our recent laboratory study (Park and Jang, 2016) to determine the kinetic rate constants that are needed to develop the AMAR model. The indoor chamber operation has been reported previously (Park and Jang 2016) (Also see Sect. S1). The indoor chamber data are listed in Table 1. The outdoor chamber experiments

were performed in the UF–APHOR dual chambers (52 m$^3$ for each chamber) to test the suitability of AMAR model to ambient condition. The light irradiation of the indoor–UV light and the sunlight are shown in Fig. S1. The detail description of the operation of outdoor chamber are also described in Sect. S1. The outdoor experimental condition for $SO_2$ heterogeneous reaction in the presence of mineral dust particles are listed in Table 2.





## 2.2 Light absorption of ATD particles

The absorbance spectrum of ATD particles was measured to develop the reaction rate constants in the kinetic model. The detailed procedure for light absorption measurement of particle samples can be found in the previous study (Zhong and Jang, 2011). The particle size distribution of ATD is shown in Fig. S2. The suspended dust particles were sampled on a Teflon coated glass fiber filter for 20 minutes. The masses difference of dust sample was measured using a microbalance (MX5, Mettler Toledo, Columbus, OH). The light absorbance of the dust filter sample ($Abs_{ATD}$) was measured using a Perkin–Elmer Lambda 35 UV–visible spectrophotometer equipped with a Labsphere RSA–PE–20 diffuse–reflectance accessory. The absorbance spectrum was normalized by particle mass and calculated to mass absorbance cross section (See Sect. S1 in Supporting Information). The resulting absorbance cross section and quantum yield of ATD dust are shown in Fig. S3.

## 3 AMAR model description

The overall schematic of the AMAR model is shown in Fig 1. In the model, the total sulfate mass concentration ($[SO_4^{2-}]_T$, μg m$^{-3}$) is predicted from the reactions in three phases: the sulfate formed in the gas phase ($[SO_4^{2-}]_{gas}$, μg m$^{-3}$), the sulfate from the aqueous phase ($[SO_4^{2-}]_{aq}$, μg m$^{-3}$) and the sulfate from dust–driven chemistry ($[SO_4^{2-}]_{dust}$, μg m$^{-3}$). The key components of the model consist of the partitioning process and the kinetic mechanisms in three phases.

(1) The gaseous inorganic species (e.g. $SO_2$, $NO_x$ and ozone) are partitioned onto both inorganic–salt (sulfuric acid and its salts) seeded aqueous particles and mineral dust particles. The gas–particle partitioning processes were treated by the adsorption–desorption kinetic mechanism.

(2) $SO_2$ oxidation in the gas phase is simulated using mechanisms previously reported in the literature (Byun and Schere, 2006; Sarwar et al., 2013; Sarwar et al., 2014; Binkowski and Roselle, 2003) (Table. S1).

(3) The partitioned $SO_2$ is heterogeneously oxidized in the inorganic–salt seeded aqueous phase based on the previously reported mechanisms (Liang and Jacobson, 1999).

(4) The formation of sulfate ($[SO_4^{2-}]_{dust}$) in the dust phase is approached using two kinetic sub–modules: the production of sulfate ($[SO_4^{2-}]_{auto}$, μg m$^{-3}$) by autoxidation in open air and sulfate formation ($[SO_4^{2-}]_{photo}$, μg m$^{-3}$) by photocatalytic reactions.





The rate constants associated with various reaction mechanisms in the AMAR model were determined by simulating indoor chamber data obtained from controlled experimental conditions (Table 1). The simulation of chamber data using the model was performed using a kinetic solver (Morpho) (Jeffries, 1998). In these mechanisms, the symbols "g", "aq", and "d" denote the

chemical species in the gas phase, inorganic–salt seeded aqueous phase, and dust phase, respectively. The unit of concentration of chemical species is molecule per $cm^3$ of air. In the following sections, the components of the AMAR model are described in detail.

### 3.1 $SO_2$ oxidation in gas phase and aerosol aqueous phase

#### 3.1.1 Gas phase oxidation

The oxidation of $SO_2$ in the gas phase has been extensively studied by numerous researchers (Baulch et al., 1984; Kerr, 1984; Atkinson and Lioyd, 1984; Calvert, 1984; Graedel, 1977; Atkinson et al., 1989). In this study, the oxidation of $SO_2$ is described using comprehensive reaction mechanisms shown in Table S1. The mechanisms can also be simplified as follows:

$$SO_2(g) + OH \rightarrow HOSO_2 \tag{R1}$$

$$HOSO_2 + O_2 \rightarrow SO_3 + HO_2 \tag{R2}$$

$$SO_3(g) + H_2O(g) + M \rightarrow H_2SO_4(aq) + M \tag{R3}$$

$$HOSO_2 + OH(g) + M \rightarrow H_2SO_4(aq) + M \tag{R4}$$

#### 3.1.2 Gas–aerosol partitioning

$SO_2$ is dissolved into hygroscopic sulfuric acid ($H_2SO_4$), which is formed in the gas phase,

*via* a partitioning process and reacts with the aqueous phase oxidants (e.g. $H_2O_2$ and $O_3$) to heterogeneously form $H_2SO_4$. The chemical species that were treated by the partitioning process include $SO_2$, $NO_x$, $O_3$, OH, $HO_2$, $H_2O_2$, HCOOH, $CH_3OOH$, $HNO_3$, $CH_3O_2$, HONO, $CH_3COOH$, and HCHO. In the model, the partitioning process is approached using the gas–particle partitioning coefficient $K_{aq,SO_2}$ ($m^3$ $\mu g^{-1}$) based on aerosol mass concentration. $K_{aq,SO_2}$ is derived

from Henry's law constant ($K_H$, mol $L^{-1}$ $atm^{-1}$) (Chameides, 1984),

$$K_{aq,SO_2} = \frac{K_{H,SO_2}RT}{\rho_{aq}} \tag{1}$$

where $R$ is the ideal gas constant (J $K^{-1}$ $mol^{-1}$) and $\rho_{aq}$ (g $cm^{-3}$) is the density of the particle, which is calculated using inorganic thermodynamic model (E–AIM II) (Clegg et al., 1998; Wexler and





Clegg, 2002; Clegg and Wexler, 2011) based on humidity and inorganic composition. The absorption–desorption process of $SO_2$ on inorganic aerosol ($In_{aq}$) is expressed as,

$$SO_2(g) + In_{aq} \rightarrow SO_2(aq) + In_{aq} \qquad k_{abs,SO_2,aq}(m^3\ m^{-2}\ s^{-1}) \quad (R5)$$

$$SO_2(aq) \rightarrow SO_2(g) \qquad k_{des\_SO_2,aq}(s^{-1}) \qquad (R6)$$

$k_{abs,SO_2,aq}$ ($s^{-1}\ m^3\ m^{-2}$) and $k_{des,SO_2,aq}$ ($s^{-1}$) are the absorption rate constant and the desorption rate constant, respectively, and are calculated as follows,

$$k_{abs,SO_2,aq} = f_{abs,aq} \frac{\omega_{SO_2} f_{aq,M2S}}{4} \qquad (2)$$

$$k_{des,SO_2,aq} = \frac{k_{abs,SO_2,aq}}{K_{aq}} \qquad (3)$$

where $f_{aq,M2S}$ ($5 \times 10^{-4}$) is the coefficient to convert the aerosol mass concentration ($\mu g\ m^{-3}$) to

the surface area concentration ($m^2\ m^{-3}$) for particle size near 100 nm and $f_{abs,aq}$ is the coefficient for absorption process. $\omega_{SO_2}$ is the mean molecular velocity ($m\ s^{-1}$) of $SO_2$ and can be calculated as follows,

$$\omega_{SO_2} = \sqrt{\frac{8RT}{\pi MW}} \qquad (4)$$

where $MW$ is molecular weight ($kg\ mol^{-1}$). In general, the characteristic time (s) of a partitioning

process ranges from $10^{-4}$ s to $10^{-2}$ s (Freiberg and Schwartz, 1981) and is much faster than both gas phase reaction and the aerosol phase reaction. In our model, $f_{abs,aq}$ was set at $2 \times 10^4$ in Eq. (2) to have fast partitioning process. The estimated characteristic time of absorption is $10^{-3}$ s. The mass concentration ($\mu g\ m^{-3}$) of inorganic seeded aqueous phase above the efflorescent relative humidity (ERH) is also dynamically calculated for the $SO_4^{2-}$–$NH_4^+$–$H_2O$ system. Colberg et al.

(2003) semiempirically predicted ERH by fitting to the experimental data based on the ammonia–to–sulfate ratio in the $SO_4^{2-}$–$NH_4^+$–$H_2O$ system. AMAR model utilizes these parameterizations to predict ERH dynamically. Ammonia is inevitable in our chamber study and mainly acts as a carryover for sulfate from previous chamber experiments. Thus, $H_2SO_4$ is fully or partially neutralized by ammonia.

### 25   3.1.3 Aerosol aqueous phase reaction

The AMAR model implements the aqueous–phase chemistry that occurs in inorganic salted aqueous aerosol (no dust) to form $SO_4^{2-}(aq)$ and $NO_3^-(aq)$. We employed the preexisting aqueous–phase kinetic reactions involving $SO_2$ (Liang and Jacobson, 1999) and $NO_x$ chemistry





(Liang and Jacobson, 1999; Hoyle et al., 2016). Thus, our simulation inherits all the possible uncertainties embedded in the original kinetic data.

The $SO_2$ dissolved in the aqueous phase is hydrolyzed into $H_2SO_3$ and dissociates to form ionic species ($HSO_3^-$ and $SO_3^{2-}$). $SO_4^{2-}(aq)$ is formed by reactions of the sulfur species in oxidation
state IV ($S(IV)(aq)$) with $OH(aq)$, $H_2O_2(aq)$, or $O_3(aq)$ (Table S1). The dissolved HONO can also dissociate to form $NO_2^-(aq)$ and result to $NO_3^-(aq)$. Each chemical species in $S(IV)(aq)$ has a different reactivity for oxidation reactions. The distribution of chemical species is affected by aerosol acidity, which is controlled by humidity and inorganic composition. Hence, the formation of sulfate is very sensitive to aerosol acidity. For example, most of the S(IV) is consumed by $H_2O_2$
at pH<4, whereas most of it is consumed by $O_3$ at pH>4. Some strong inorganic acids, such as sulfuric acid, influence aerosol acidity. In AMAR, aerosol acidity ($[H^+]$, mol $L^{-1}$) is predicted using the inorganic thermodynamic model E–AIM II (Clegg et al., 1998; Wexler and Clegg, 2002; Clegg and Wexler, 2011) based on the ammonia–to–sulfate ratio and RH. When the ammonia–to–sulfate ratio is greater than 0.8, the prediction of $[H^+]$ is corrected based on the method of Li and
Jang (2012). At high $NO_x$ levels, $NO_2^-(aq)$ competes with $S(IV)(aq)$ for the reaction with $OH(aq)$, $O_3$, or $H_2O_2$ (Table S1)(Ma et al., 2008). However, the HONO concentration becomes high at high $NO_x$ levels and enhances $SO_2$ oxidation in the inorganic–salt seeded aqueous phase due to the formation of OH radicals *via* photolysis of HONO.

### 3.2 Heterogeneous oxidation in the presence of mineral dust particles

The heterogeneous chemistry in the presence of dust particles has been newly established in the AMAR model. The dust phase module consists of a partitioning process (Sect. 3.2.1) and heterogeneous chemistry for $SO_2$ and other trace gases (ozone, HONO, and $NO_2$) (Table 3) (Fig. 1). The heterogeneous chemistry of $SO_2$ is handled by autoxidation (Sect. 3.2.2) and photooxidation under UV light (Sect. 3.2.4). In dust–phase photochemistry, the central mechanism
for $SO_2$ oxidation is operated by the surface oxidants (e.g. OH(d)), which is generated *via* the photoactivation process of semiconductive metal oxides in dust particles (Sect. 3.2.3).

### 3.2.1 Gas–dust particle partitioning

The gas–dust partitioning constant ($K_{d,SO_2}$, $m^3$ $m^{-2}$) of $SO_2$ is defined as,





$$K_{d,SO_2} = \frac{[SO_2]_d}{[SO_2]_g A_{Dust}} \quad (m^3\ m^{-2}) \tag{5}$$

$A_{dust}$ ($m^2\ m^{-3}$) is the geometric surface concentration of ATD dust particles and is calculated by multiplying the dust mass concentration ($\mu g\ m^3$) by a geometric surface–mass ratio ($f_{dust,M2S}$) of ATD particles ($3.066 \times 10^{-6}$, $m^2\ \mu g^{-1}$). The $SO_2$ absorption and desorption processes for the dust

phase are expressed as

$$SO_2(g) + A_{Dust} \rightarrow SO_2(d) + A_{Dust} \quad k_{abs\_SO_2,dust}(m^3\ m^{-2}\ s^{-1}) \tag{R7}$$

$$SO_2(d) \rightarrow SO_2(g) \quad\quad\quad\quad k_{des\_SO_2,dust}(s^{-1}) \tag{R8}$$

$k_{abs\_SO_2,dust}$ ($s^{-1}\ m^3\ m^{-2}$) and $k_{des\_SO_2,dust}$ ($s^{-1}$) are the absorption rate constant and the desorption rate constant, respectively. At equilibrium, the absorption rate (R7) equals the desorption rate

(R8). Thus, $K_{d,SO_2}$ can be expressed as

$$K_{d,SO_2} = \frac{k_{abs\_SO_2,dust}}{k_{des\_SO_2,dust}} \quad (m^3\ m^{-2}) \tag{6}$$

$K_{d,SO_2}$ is set at 1.63 ($m^3\ m^{-2}$, at 298K for dry particles) based on the literature data (Adams et al., 2005; Huang et al., 2015). The characteristic time to reach to equilibrium is very short (Sect. 3.1.1). In kinetic mechanisms, $k_{abs\_SO_2,dust}$ was set at $1.7 \times 10^3\ s^{-1}\ m^3\ m^{-2}$ for dry particles (20%

RH) using the same approach as Eq. (2). The resulting characteristic time for $k_{abs\_SO_2,dust}$ is $10^{-6}$ s. The characteristic time of the reaction of $SO_2$ with an OH radical ($10^6$ molecules $cm^{-3}$) is about $10^6$–$10^7$ s in gas phase and $10^5$–$10^6$ s in both aqueous phase and dust phase.

To consider the effect of temperature on $K_{d,SO_2}$, the temperature dependency of $k_{des\_SO_2,dust}$ (Eq. (6)) is derived from the Henry's constant (Chameides, 1984). $K_{d,SO_2}$ (Eq. (5)) is

also influenced by aerosol water content (Zuend et al., 2011) as well as the dissociation of $H_2SO_3$, which is operated by aerosol acidity ([$H^+$]) and an acid dissociation constant ($Ka_{SO_2}$)(Martell and Smith, 1976). Thus, $k_{des\_SO_2,dust}$ is expressed as,

$$k_{des\_SO_2,dust} = 2 \times 10^9 \exp\left(-\frac{3100}{T}\right) / \left(F_{water}\left(1 + \frac{Ka_{SO_2}}{[H^+]}\right)\right) \quad (s^{-1}) \tag{7}$$

$Ka_{SO_2}$ is 0.013 (mol $L^{-1}$) at 298K (Martell and Smith, 1976). The influence of the dissociation of

inorganic acid on $K_{d,SO_2}$ is accounted for by the term $\left(1 + \frac{Ka_{SO_2}}{[H^+]}\right)$ in Eq. (7). The estimation of [$H^+$] is treated in the same ways as aqueous chemistry (Sect. 3.1.3). $F_{water}$, a numeric number, was introduced into the model to estimate the water fraction of total dust particles. The hygroscopic property of mineral dust dynamically changes because dust can be substantially modified by direct





reaction of some of its components (e.g. $CaCO_3$) with inorganic acids such as $H_2SO_4$ and $HNO_3$. When dust forms $Ca(NO_3)_2$, dust becomes more hygroscopic. Nitrate salts deliquesce at very low RH (17%) (Krueger et al., 2003; Krueger et al., 2004; William et al., 2005). $CaSO_4$ is, however, relatively hydrophobic. Nitrate salts exist only when sulfate concentrations is very low. $F_{water}$

originated from the hygroscopic property of indigenous dust (first term in Eq. (8)), the inorganic nitrates formed from the reaction of adsorbed $HNO_3$ with dust (second term), the inorganic sulfate ($SO_4^{2-}$–$NH_4^+$–$H_2O$ system, third term).

$$F_{water} = \exp(4.4RH) + 3.7\exp(4.4RH)\frac{[NO_3^-]}{[Dust]} + \frac{M_{in,water}}{[Dust]} \qquad (8)$$

$M_{in,water}$ is the water concentration ($\mu g \; m^{-3}$) associated with inorganic sulfate and calculated using

E–AIM II. Both $[NO_3^-]$ and $M_{in,water}$ are normalized by the mass concentration of ATD particles ([Dust], $\mu g \; cm^{-3}$). $F_{water}$ is first determined using chamber simulation of $SO_2$ heterogeneous oxidation (first and third terms in Eq. (8)) (D1, D2 and D3 in Table 1) under varied RH levels and extended to $SO_2$ oxidation in the presence of $NO_x$ (L6 and L7 in Table 1). Among temperature, RH and aerosol acidity, the most influential variable is RH due to the variation of $F_{water}$ (see

sensitivity analysis in Sect. 5).

### 3.2.2 Autoxidation of $SO_2$ on dust surface

Typically, autoxidation of $SO_2$ is an oxidation process *via* the reaction of adsorbed $SO_2$ (R7 and R8) with an oxygen molecule. In the model, $[SO_4^{2-}]_{auto}$ is defined as the sulfate resulted from any oxidation reactions (autoxidation in open air and oxidation with ozone) of $SO_2$ without

UV light (Fig. 1). In autoxidation, the reaction of $SO_2(d)$ with the oxygen molecules is treated as the first order reaction (assuming the concentration of oxygen is constant as $2 \times 10^5$ ppm).

$$SO_2(d) \xrightarrow{O_2(g)} SO_4^{2-}(d) \qquad k_{auto} = 5 \times 10^{-6} \; (s^{-1}) \qquad (R9)$$

The $SO_2$ autoxidation reaction rate constant ($k_{auto}$, $s^{-1}$) is semiempirically determined by simulating experimental data (D1, D2 and D3 in Table 1). For comparison with other studies, we estimate

the reactive uptake coefficient ($\gamma_{SO_4^{2-},auto}$) of $SO_2$ onto ATD dust in the absence of ozone and $NO_x$ (Fig. 2).

$$\gamma_{SO_4^{2-},auto} = \frac{4K_{d,SO_2}k_{auto}}{\omega_{SO_2}} \qquad (9)$$

$\gamma_{SO_4^{2-},auto}$ is proportional to $K_{d,SO_2}$, and influenced by humidity (Eq. (7)).



### 3.2.3 Photoactivation of dust particles and heterogeneous formation of OH radicals

The reactive uptake of $SO_2$ on particles is traditionally treated as a first order process (Ullerstam et al., 2003; Li et al., 2007). Such an approach is appropriate for simple autoxidation mechanisms, but not for the complex heterogeneous photooxidation of $SO_2$. In the AMAR model,

the heterogeneous photooxidation of $SO_2$ is approached in three steps: (1) the formation of an $e^-_{cb}$–$h^+_{vb}$ pair *via* photoactivation of dust particles, (2) the formation of OH(d) *via* the reaction of an $e^-_{cb}$–$h^+_{vb}$ pair with a water or oxygen molecule, and (3) the reaction of adsorbed $SO_2$ with the resulting OH(d) (second–order reactions) (Table S1).

The photoactivation of dust particles and the recombination reaction of an electron–hole

pair (e_h) are added into the model.

$$\text{Dust} \xrightarrow{h\upsilon} \text{Dust} + \text{e\_h} \qquad k^j_{e\_h} = j_{[ATD]} \qquad\qquad \text{(R10)}$$

$$\text{e\_h} \to \text{energy} \qquad k_{recom} = 1\times10^{-2}\ (\text{s}^{-1}) \qquad\qquad \text{(R11)}$$

where $k^j_{e\_h}$ is the photoactivation rate constant to form $e^-_{cb}$–$h^+_{vb}$ pairs and $k_{recom}$ is the reaction rate constant of recombination (heat radiation) of an electron and a hole. The value of $k_{recom}$ is

set at a large number to prevent the accumulation of electron–hole pairs. The formation of OH(d) is expressed as

$$\text{e\_h} + O_2(g) \to \text{OH(d)} \qquad k_{OH,O_2} = 1\times10^{-22}\exp(2.3RH)\ (\text{cm}^3\ \text{molecules}^{-1}\ \text{s}^{-1})\ \text{(R12)}$$

$k_{OH,O_2}$ is the reaction rate constant to form OH(d) and is estimated as a function of humidity using indoor chamber data (L2, L3 and L4 in Table 1).

In R10, $k^j_{e\_h}$ is the operational rate constant for the photoactivation of dust particles and is dependent on the photolysis rate constant, $j_{[ATD]}$ ($\text{s}^{-1}$). Like the typical photolysis of a gaseous molecule, the photocatalytic production of the $e^-_{cb}$–$h^+_{vb}$ pair is linear to both the actinic flux ($I(\lambda)$, photons $\text{cm}^{-2}\ \text{nm}^{-1}\ \text{s}^{-1}$) originating from the light source and the photocatalytic property of dust particles. The value of $j_{[ATD]}$ is determined by $I(\lambda)$, the absorption cross section ($\sigma(\lambda)$, $\text{cm}^2\ \mu\text{g}^{-1}$),

and the quantum yield ($\phi(\lambda)$) of dust conducting matter at each wavelength range ($\lambda$, nm),

$$j_{[ATD]} = \int_{\lambda 1}^{\lambda 2} I(\lambda)\sigma(\lambda)\phi(\lambda)d\lambda \qquad\qquad \text{(10)}$$

In the model, $\sigma(\lambda)$ is the light absorption needed to activate dust–phase semiconducting metal oxides (excitation from a ground energy level to a conducting band), and $\phi(\lambda)$ is the probability of yielding the $e^-_{cb}$–$h^+_{vb}$ pair in the dust phase. Both $\sigma(\lambda)$ and $\phi(\lambda)$ cannot be directly measured because



of complexity in the quantity of photoactive conducting matter in dust particles and the irradiation processes of the $e^-_{cb}$–$h^+_{vb}$ pair. In order to deal with $\sigma(\lambda) \times \phi(\lambda)$, we calculated the mass absorption cross section of dust particles ($MAC_{ATD}$, $m^2$ $g^{-1}$), which was determined using the absorption coefficient of ATD particles ($b_{ATD}$, $m^{-1}$) with the particle concentration ($m_{ATD}$, g $m^{-3}$):

$$MAC_{ATD} = \frac{b_{ATD}}{m_{ATD}} \qquad (11)$$

In Eq. (11), $b_{ATD}$ can be calculated from the absorbance of dust filter sample ($Abs_{ATD}$, dimensionless) measured using a reflective UV–visible spectrometer (Fig. S3):

$$b_{ATD} = \frac{Abs_{ATD} \, A}{f \, V} \ln(10) \qquad (12)$$

where A = $7.85 \times 10^{-5}$ ($m^2$) is the sampled area on the filter and $V$ ($m^3$) is the total air volume

passing through the filter during sampling. In order to eliminate the absorbance caused by filter material scattering, a correction factor ($f = 1.4845$) is obtained from a previous study (Zhong and Jang, 2011) and coupled into Eq. (12). The preliminary study showed that the effect of aerosol scattering on the $b_{abs}$ values of the aerosol collected on the filter was negligible. Further, Bond (2001) reported that particle light scattering does not significantly influence spectral absorption

selectivity. The $MAC_{ATD}$ of dust particles originates from photocatalytic conducting matter (e.g. $TiO_2$) as well as light–absorbing matter (e.g. gypsum and metal sulfate). Thus, the $MAC_{ATD}$ spectrum is adjusted using the known $TiO_2$ absorption spectrum (Reyes–Coronado et al., 2008) and applied to $\sigma(\lambda) \times \phi(\lambda)$ (Fig. S3). The resulting $\sigma(\lambda) \times \phi(\lambda)$ spectrum is applied to Eq. (10) to calculate $j_{[ATD]}$ (R10).

**3.2.4 Heterogeneous photooxidation of SO₂**

SO₂ is oxidized by OH(d) on the surface of ATD particles as follows,

$SO_2(d) + OH(d) \rightarrow SO_4^{2-}(d)$ $\qquad k_{photo} = 1.0 \times 10^{-12}$ ($cm^3$ molecule$^{-1}$ s$^{-1}$) (R13)

where $k_{photo}$ is the reaction rate constant of SO₂ with OH(d) and is estimated from gas phase reaction (R1). Combining Eq. (4), Eq. (5), R11 and R15, the reactive uptake coefficient

($\gamma_{SO_4^{2-},photo}$) of SO₂ on ATD particles under UV light can be written as,

$$\gamma_{SO_4^{2-},photo} = \frac{4K_{d,SO2}\left(k_{photo}[\text{OH(d)}] + k_{auto}\right)}{\omega_{SO_2}} \qquad (13)$$

$\gamma_{SO_4^{2-},photo}$ is the constant at a given concentration of OH(d) (for a given light source, dust concentration, and humidity) (R10 and R12). Figure 2 illustrates $\gamma_{SO_4^{2-},photo}$ values at three





different RHs, which were obtained using indoor chamber data. $\gamma_{SO_4^{2-},photo}$ is significantly influenced by both UV light and humidity. For example, $\gamma_{SO_4^{2-},photo}$ is one order of magnitude higher than $\gamma_{SO_4^{2-},auto}$ at low $NO_x$ levels (<5 ppb), and $\gamma_{SO_4^{2-},photo}$ increased from $2.0 \times 10^{-5}$ to $1.24 \times 10^{-4}$ when the RH changed from 20% to 80%.

### 3.3 Impact of ozone and $NO_x$ on heterogeneous chemistry of $SO_2$

To date, most studies of the effect of $NO_x$ on sulfate formation have been limited to the reaction in dark condition. For example, previous laboratory studies using pure metal oxides reported the acceleration of the heterogeneous oxidation of $SO_2$ by $NO_x$ in dark conditions (Ma et al., 2008; Liu et al., 2012). For the effect of ozone, the recent chamber study by Park and Jang (2016) showed significant enhancement of heterogeneous photooxidation of $SO_2$. In the AMAR model, the formation of sulfate is also modulated by the involvement of ozone and $NO_x$ in both autoxidation and photochemistry on the surface of dust particles (Fig. 1).

#### 3.3.1 Dust–phase ozone chemistry

Like $SO_2$ (R5 and R6), the gas–dust partitioning coefficient of ozone, $K_{d,O_3}$, is determined using the value found in the literature (Michel et al., 2003; Ullerstam et al., 2002):

$$K_{d,O_3} = 1.9 \times 10^{-6} \exp\left(\frac{2700}{T}\right) \quad (\text{m}^3 \text{ m}^{-2}, 20\% \text{ RH}) \qquad (14)$$

The partitioning process is also treated by the adsorption–desorption kinetic mechanism as shown in R7 and R8 (Table 3: partitioning). Ozone can decay catalytically in the dust phase, forming an oxygen molecule and surface–bound atomic oxygen (Usher et al., 2003; Chang et al., 2005). The formed atomic oxygen reacts with $SO_2$(d) to form sulfate (Ullerstam et al., 2002; Usher et al., 2002):

$$SO_2(d) + O_3(d) \rightarrow SO_4^{2-}(d) + O_2 \quad k_{auto,O_3} = 2 \times 10^{-11} \ (\text{cm}^3 \text{ molecules}^{-1} \text{ s}^{-1}) \quad (R14)$$

$k_{auto,O_3}$ is estimated using indoor chamber data (D4 in Table 1). In the presence of 200 µg m$^3$ of ATD particles and 30 ppb of ozone, the concentration of $O_3$(d) is estimated as $9.4 \times 10^6$ molecule cm$^{-3}$. Under this condition, the characteristic time of the autoxidation by ozone (R14) is $5 \times 10^3$ s and is much faster than the autoxidation by oxygen (R9, $2 \times 10^5$ s). At nighttime, in the presence of ozone, the autoxidation of $SO_2$(d) yields a significant amount of sulfate.





Under UV light, ozone is also involved in the production of the surface oxidants ($O_3^-$, $HO_3\cdot$ and OH) that further promote heterogeneous oxidation of $SO_2$. $O_3$(d) acts as an acceptor for $e^-_{cb}$–$h^+_{vb}$ and forms OH(d):

$$e\_h + O_3(d) \rightarrow \cdot OH(d) + O_2 \quad k_{OH,O_3}=1\times10^{-12} \text{ (cm}^3 \text{ molecules}^{-1} \text{ s}^{-1}) \text{ (L5 in Table 1)(R15)}$$

### 3.3.2 Dust–phase $NO_x$ chemistry

The equilibrium constant of $NO_2$ on the dust surface ($K_{d, NO_2}$) is scaled from $K_{d,SO_2}$ using the known Henry's law constant for $SO_2$ ($K_{H,SO_2}$, Eq. (1)) and $NO_2$ ($K_{H,NO_2}$) (Chameides, 1984), and written as:

$$K_{d, NO_2} = K_{d, SO_2} \frac{K_{H,NO_2}}{K_{H,SO_2}} = 3.7\times10^{-6}\exp(\frac{2500}{T}) \text{ (m}^3 \text{ m}^{-2}, 20\% \text{ RH)} \tag{15}$$

The adsorbed $NO_2$ first reacts with $e^-_{cb}$(d) or $\cdot O_2^-$(d) on the dust surface (R10) and forms HONO(d) (Ma et al., 2008; Liu et al., 2012; Saliba and Chamseddine, 2012; Saliba et al., 2014). In AMAR, the formation of HONO(d) is simplified into:

$$e\_h + NO_2(d) \rightarrow HONO(d) \quad k_{NO2}=6\times10^{-12} \text{ (cm}^3 \text{ molecules}^{-1} \text{ s}^{-1}) \text{ (L7 and L8 in Table 1) (R16)}$$

HONO(d) is further decomposed through photolysis and yields OH(d):

$$HONO(d) \xrightarrow{hv} \cdot OH(d) + NO \quad k^j_{HONO} = j_{[HONO]} \text{ (s}^{-1}) \tag{R17}$$

The photolysis rate constant of HONO(d) is treated with the one for gaseous HONO ($j_{[HONO]}$). Similar to autoxidation of $SO_2$ (Sect. 3.2.2), $NO_2$(d) autoxidizes to form nitrate:

$$NO_2(d) \xrightarrow{O_2(g)} NO_3^-(d) \qquad k_{auto,NO_2} = 6\times10^{-5} \text{ (s}^{-1}) \tag{R18}$$

$NO_2$ reacts with OH(d):

$$NO_2(d) + OH(d) \rightarrow NO_3^-(d) \quad k_{photo,NO_2} = 1\times10^{-10} \text{ (cm}^3 \text{ molecules}^{-1} \text{ s}^{-1}) \tag{R19}$$

$k_{auto,NO_2}$ and $k_{photo,NO_2}$ was determined using the simulation of indoor chamber data (L7 and L8 in Table 1). The estimation of the gas–dust partitioning coefficients of HONO ($K_{d,HONO}$) (Becker et al., 1996) and $HNO_3$ ($K_{d, HNO_3}$) (Schwartz and White, 1981) was approached using the similar method for $SO_2$ (Table 3). $N_2O_5$ forms nitrate *via* a reactive uptake process as shown in Table 3 (reaction 11).





## 4 Simulation of AMAR model under ambient sunlight

At the beginning of the development of the AMAR model, the kinetic parameters to predict the formation of sulfate and nitrate in the presence of ATD particles were leveraged using an indoor chamber. In order to test the feasibility of the resulting AMAR model, the UF–APHOR data using

natural sunlight were simulated (Table 2). The chamber dilution (measured by $CCl_4$) and the wall process of gaseous compounds (e.g. ozone, $SO_2$, HONO, $NO_2$) and particles were integrated with the kinetic mechanisms to simulate UF–APHOR data (Sect. S1). As shown in Fig. 1, the model inputs are the concentration of chemical species, the amount of dust, and the meteorological variables that are commonly found at regional scales. The dual chambers allow for two controlled

experiments to be performed simultaneously under the same meteorological conditions.

### 4.1 Simulations for different dust loadings

Figure 3 shows that the predicted $[SO_4^{2-}]_T$ is in good agreement with experimental observations, which were performed under low $NO_x$ conditions ($NO_x < 5$ ppb) for two different dust loadings as well as two different $SO_2$ levels. The greater increase in $[SO_4^{2-}]_T$ appeared with

the higher sunlight intensity (between 11 AM and 2 PM). In Fig. 3(a), the predicted $[SO_4^{2-}]_T$ increased by 63% (at 3 PM) with 290 µg m$^{-3}$ of ATD particles compared to the $[SO_4^{2-}]_T$ without dust particles. Figure 3(b) confirms that the larger dust particle loading yields more $[SO_4^{2-}]_T$. In Fig. 3(c), $[SO_4^{2-}]_T$ was predicted with high and low initial concentrations of $SO_2$ for a given dust loading. The time profiles of the simulation of concentrations of $NO_x$, ozone, $SO_2$ and dust are

shown in Fig. S4.

Because of the large size of dust particles, the wall processes (e.g. settling and wall deposition) of dust particles is greater than that of the sulfate particles originated from $[SO_4^{2-}]_{aq}$ (no dust). Hence, the fraction of $[SO_4^{2-}]_{dust}$ to $[SO_4^{2-}]_T$ declines over the course of the chamber experiment. To estimate how the predicted $[SO_4^{2-}]_T$ is attributed to $[SO_4^{2-}]_{aq}+[SO_4^{2-}]_{gas}$ (non–dust

sulfate) and $[SO_4^{2-}]_{dust}$ without wall processes, Fig. 3(d), 3(e), and 3(f) are reconstructed from Fig. 3(a), 3(b), and 3(c), respectively. As shown in the inner pie chart of Fig. 3(d), a significant fraction of $[SO_4^{2-}]_T$ is attributed to $[SO_4^{2-}]_{dust}$ (0.73). Only the 0.03 of $[SO_4^{2-}]_T$ originates from autoxidation. In Fig. 3(e), the fraction of $[SO_4^{2-}]_{dust}$ to $[SO_4^{2-}]_T$ increases from 0.48 to 0.85 with the increase of dust loading from 90 µg m$^{-3}$ to 403 µg m$^{-3}$. The increased dust loading promotes both the

adsorption of $SO_2$ onto dust particles and the production of dust–phase oxidants, and thus yields





more sulfate production. With the increase of the initial concentration of $SO_2$ from 119 ppb to 272 ppb in Fig. 3(f), the $[SO_4^{2-}]_{dust}$ fraction decreases from 0.5 to 0.33, although $[SO_4^{2-}]_T$ increases from 17.6 $\mu g\ m^{-3}$ to 28.2 $\mu g\ m^{-3}$. The high concentration of $SO_2$ increases sulfate formation in the gas phase and subsequentially promotes aqueous phase chemistry. The sulfuric acid formed in the

aqueous phase is hydrophilic and creates a positive feedback loop which aggravates the growth of aqueous aerosol. Overall, the variation in dust concentration is more influential on $[SO_4^{2-}]_T$ than that of $SO_2$.

### 4.2 Simulation of $NO_x$ effect

Figure 4 shows that the model performs well in predicting $[SO_4^{2-}]_T$ in various levels of
$NO_x$. Both the model–predicted and observed $[SO_4^{2-}]_T$ are lower at higher $NO_x$. The concentrations of background hydrocarbon in Fig. 4(a) and Fig. 4(b) are low (no additional hydrocarbons, Sect. S1), while 30 ppb of isoprene were injected in Fig. 4(c) to increase the amount of ozone. Isoprene oxidation in the gas phase was simulated using MCM (version 3.3.1) (Jenkin et al., 1997; Saunders et al., 2003). Figure 4(d) is reconstructed from Fig. 4(a), 4(b) and 4(c) to illustrate how $[SO_4^{2-}]_T$ is
attributed to the aqueous–phase reaction ($[SO_4^{2-}]_{gas}+[SO_4^{2-}]_{aq}$), dust–phase autoxidation ($[SO_4^{2-}]_{auto}$), and dust photochemistry ($[SO_4^{2-}]_{photo}$). $[SO_4^{2-}]_{photo}$ is significantly suppressed at high $NO_x$ levels because $NO_2$ competes with $SO_2$ to consume OH radicals. Both $[SO_4^{2-}]_{auto}$ and $[SO_4^{2-}]_{photo}$ are noticeably large in the columns reconstructed from Fig. 4(c) because of the contribution of ozone to the heterogeneous chemistry of dust (R14 and R15). The simulated concentrations of
$NO_x$, ozone, $SO_2$ and dust are shown in Fig. S5.

The time profiles of the predicted $[NO_3^-]_T$ are also shown in Fig. 4(a), 4(b), and 4(c). In the morning, $NO_2$ quickly oxidizes to accumulate nitric acid in the dust phase. The dust–phase nitric acid might rapidly react with alkaline carbonates (e.g. K, Na, Ca and Mg ions) in the dust phase and form nitrate salts ($NO_3^-$(d_salt) in reaction 12 in Table 3). As described in Sect. 3.2.1,
these nitrate salts are very hygroscopic and further enhance gas–dust partitioning of gaseous species including $HNO_3$, $SO_2$, and HONO at high humidity (in the morning). With increasing sunlight intensity, the temperature increases but humidity decreases (20%, Fig. S6). In addition to meteorological conditions, the formation of low–volatility sulfuric acid can deplete nitrate *via* evaporation of volatile nitric acid ($SO_4^{2-}$(d_salt) in reaction 13 and 14 in Table 3) from the dust
surface. The capacity of ATD particles to form nitrate salts (or sulfate salts) is limited by the





amount of carbonates and metal oxides on the surface of dust particles. This capacity is estimated to be 1.7 ppb per 100 μg of dust, which was determined by comparing the actual aerosol acidity, as measured by the colorimetry integrated with a reflectance UV–visible spectrometer (C–RUV), to the aerosol acidity predicted by the inorganic thermodynamic model (E–AIM II) using the

inorganic composition from PILS–IC (Li et al., 2015; Beardsley and Jang, 2016). As shown in Fig. 4, the effect of $HNO_3$ on the heterogeneous reaction is negligible during daytime because sulfuric acid, a strong acid, depletes partitioning of $HNO_3$ (Eq. (15)). At the end of the photooxidation, nitrate is slightly underestimated because some observed nitrate may be trapped under the layer of hydrophobic alkaline sulfate formed *via* aging of ATD particles (effloresced).

The surface HONO(d), which formed *via* the photocatalytic process of $NO_2$ (R16), can influence the production of OH(d). However, the model analysis originated from the integrated reaction rate (IRR), an accumulated flux of chemical formation, suggests that the contribution of HONO(d) to OH(d) production is relatively small compared to the direct photocatalytic process caused by dust particles shown in Sect. 3.2.3.

**5 Sensitivity and uncertainties**

The sensitivity of sulfate prediction to major variables (e.g. temperature, humidity, sunlight profile, the concentration of $SO_2$ and $NO_x$, and dust loading) is illustrated in Fig. 5. To avoid the suppression of dust chemistry at high $NO_x$ levels, the most sensitivity tests were performed at low levels of $NO_x$. The stacked chart normalized with $[SO_4^{2-}]$ in Fig. 5 shows how $[SO_4^{2-}]_T$ is attributed

to $[SO_4^{2-}]_{auto}$, $[SO_4^{2-}]_{photo}$ and $[SO_4^{2-}]_{aq}+[SO_4^{2-}]_{gas}$ (non–dust chemistry).

Figure 5(a) illustrates that the reduction of $[SO_4^{2-}]_T$ at a higher temperature (288K vs. 308K) is ascribed to the decrease in the partitioning process. Figure 5(b) shows that $[SO_4^{2-}]_T$ increases by a factor of 3.5 with RH increasing from 25% to 80%. Humidity plays an important role in the modulation of both aerosol acidity and liquid water content, and ultimately influences

the partitioning process (e.g. $SO_2$ partitioning on dust surface) and dust–phase chemistry (e.g. production of OH(d)). In the stacked column chart of Fig. 5(b), the contribution of $[SO_4^{2-}]_{dust}$ to $[SO_4^{2-}]_T$ increases from 0.64 to 0.79 with increasing RH suggesting that dust chemistry is more sensitive to humidity than aqueous phase chemistry. Figure 5(c) presents $[SO_4^{2-}]_T$ at two different sunlight intensities (winter on 22 November 2016 vs. summer on 11 May 2015) in Gainesville,

Florida (latitude/longitude: 29.64185°/-82.347883°). As shown in Fig. 5(d), with $SO_2$





concentrations increasing from 20 ppb to 100 ppb, $[SO_4^{2-}]_T$ increases by a factor of two in the given simulation condition. However, the fraction of $[SO_4^{2-}]_{dust}$ to $[SO_4^{2-}]_T$ decreased from 0.88 to 0.72. The effect of the concentration of $SO_2$ on $[SO_4^{2-}]_T$ has been discussed in Sect. 4.1 above. Figure 5(e) shows the sensitivity of $[SO_4^{2-}]_T$ to the ATD loading (100, 200, and 400 µg m$^{-3}$). Figure

5(f) illustrates how sulfate formation is suppressed by different $NO_x$ levels (also see Sect. 3.3.2).

Figure S7 illustrates the influence of the uncertainties in the major model parameters on the prediction of $[SO_4^{2-}]_T$. The uncertainty in $K_{d,SO_2}$ (±16%) of $SO_2$ was determined using a value from the literature (Adams et al., 2005). The variation in $[SO_4^{2-}]_T$ due to the uncertainty in $K_{d,SO_2}$ is as small as ±2%. The reaction rate constants of dust chemistry in the model (Table 3) were semi–

empirically determined using preexisting indoor chamber data (Park and Jang, 2016) and chamber characterization. The uncertainty in rate constants associated with observed sulfate concentrations is about ±10%. Fig. S7 also shows the variation in $[SO_4^{2-}]_T$ due to the uncertainty in both the reaction of $SO_2$ with dust–surface OH radicals ($k_{photo}$) and the production rate constant of dust– surface OH radicals ($k_{OH,O_2}$). Among $K_{d,SO_2}$, $k_{photo}$, and $k_{OH,O_2}$, the highest uncertainty appears

in $k_{OH,O_2}$.

Most simulations of sulfate in this study are limited to environmental conditions under low concentrations of hydrocarbons. In the future, the model should be evaluated for the chamber data generated from various mixes of $SO_2$, $NO_x$, and hydrocarbons in the presence of mineral dust. The inorganic thermodynamic model (e.g. E–AIM II) was employed here to estimate [H$^+$] and the

liquid water content ($M_{in,water}$) for the $SO_4^{2-}$–$NH_4^+$–$H_2O$ system (excluding $SO_4^{2-}$(d_salt) in reaction 13 of Table 3: dust phase) (Eq. (8)) in both inorganic–salt seeded aqueous phase and dust phase chemistry. The uncertainty in $M_{in,water}$ and [H$^+$] influences partitioning of $SO_2$ and $NO_x$, as well as $[SO_4^{2-}]_T$. The uncertainties in the prediction of [H$^+$] using inorganic thermodynamic models are large because of the limited data (Clegg et al., 1998; Wexler and Clegg, 2002), especially for

ammonia–rich inorganic salts in the low RH range. In this study, our model uses the corrected estimation of [H$^+$] based on the filter–based C–RUV technique (Li et al., 2015). The estimated uncertainty in the C–RUV method is ±18%, and results in a ±7% variation in $[SO_4^{2-}]_T$. The dust surface area in AMAR is calculated using the geometric surface area. To extend the AMAR model to other dust materials, the molecular level surface area (BET surface area) should be considered

in the future.




## 6 Conclusion and atmospheric implication

The AMAR model of this study was developed to predict the oxidation of $SO_2$ and $NO_x$ using comprehensive kinetic mechanisms in the gas phase, inorganic seeded aqueous phase, and dust phase. The thermodynamic parameters engaged in the partitioning process between gas,

inorganic salted aqueous aerosol and dust phases were obtained from known data in the literature (Table 3), and the kinetic parameters for dust chemistry were estimated using previously reported indoor chamber data (Park and Jang, 2016). Overall, the AMAR simulations were consistent with experimentally observed outdoor chamber data (Fig. 3 and Fig. 4) under ambient sunlight. As discussed in the sensitivity analysis (Sect. 5), both the $[SO_4^{2-}]_T$ and the relative distribution of

mechanism–based sulfate formation are sensitive to all major variables (model inputs) including temperature, humidity, sunlight intensity, the quantity of dust loading, and concentrations of $NO_x$ and $SO_2$.

In order to assess the importance of dust chemistry in ambient conditions, the prediction of sulfate formation in the presence of ATD dust needs to be extended to 24 h simulations under

various environmental conditions. Figure S8 shows the output simulated for 24 h with 200 μg m$^3$ of ATD particle loading under urban (40 ppb $NO_x$; VOC/$NO_x$ < 5; 20 ppb $SO_2$) and rural atmospheres (5 ppb $NO_x$; VOC/$NO_x$ > 20; 2 ppb $SO_2$). At nighttime, when the temperature drops and humidity increases (70–90%, Fig. S6), the contribution of $[SO_4^{2-}]_{auto}$ to $[SO_4^{2-}]_T$ becomes larger than the typical chamber simulation during the daytime. In a rural environment, $[SO_4^{2-}]_{photo}$

is still the most influential on sulfate formation (0.57 fraction of $[SO_4^{2-}]_T$ in Fig. S8(a)). For the simulation in a polluted area (Fig. S8(b)), the fraction of $[SO_4^{2-}]_{photo}$ to $[SO_4^{2-}]_T$ significantly decreases (0.13) because of the suppression induced by $NO_x$ (Sect. 3.3.2), but the fraction of $[SO_4^{2-}]_{auto}$ to $[SO_4^{2-}]_T$ increases (0.34). With decreasing sunlight intensity (after 5 PM), Fig. S8 shows the rapid increases in $[SO_4^{2-}]_{auto}$ due to the reaction of dust–phase $SO_2$ with ozone, which is the

result of daytime photooxidation (Sect. 3.3.1). Fig. S8 suggests that the failure to predict sulfate formation without accurate dust chemistry ($[SO_4^{2-}]_{auto}$ + $[SO_4^{2-}]_{photo}$) can lead to substantial underestimation of the quantity of total sulfate at regional or global scales. $SO_2$ autoxidation alone may partially improve the prediction of sulfate in the presence of mineral dust, but sulfate production can still be largely underestimated and incorrectly predicted in time series when

heterogeneous photocatalytic reactions in kinetic mechanisms are not considered.



The ATD particles in this study have chemical and physical properties different from ambient mineral dust particles. In general, the uptake coefficient of $SO_2$ in authentic mineral dust particles (e.g. Gobi Desert dust and Saharan dust) is known to be higher than that of ATD particles (Crowley et al., 2010). Thus, the effect of ambient dust particles on heterogeneous photocatalytic

5    oxidation would be much more important than that of the ATD particles of this study. To extend the AMAR model to the prediction of sulfate in the presence of ambient dust particles, the model parameters related to rate constants, partitioning process, and the physical characteristics (e.g. surface area) of dust particles need to be updated with chamber data.

**Acknowledgments**

10   This work was supported by grants from the National Institute of Metrological Science (NIMS–2016–3100), the Ministry of Science, ICT, and Future Planning at South Korea (2014M3C8A5032316) and the Fulbright Scholar (from USA to Mongolia).



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





**Table 1. Experiment conditions and simulation results for SO₂ heterogeneous photooxidation on the surface of ATD particles at variety condition of humidity (RH), light sources and initial concentration of traces using indoor chamber data.**

| Exp. No.[a] | UV | RH[b] (%) | Temp.[b] (K) | ATD dust[c] ($\mu g\ m^{-3}$) | SO₂[d] (ppb) | NO/NO₂[d] (ppb) | O₃[d] (ppb) | NH₄⁺[d] ($\mu g\ m^{-3}$) | Duration[e] (min) | Exp. $[SO_4^{2-}]_T$[f] ($\mu g\ m^{-3}$) |
|---|---|---|---|---|---|---|---|---|---|---|
| | | | | | | Initial Concentration | | | | |
| D1 | Off | 21.0 | 295.9 | 295 | 267 | N.A. | N.A. | N.A. | 150 | 0.61±0.02 |
| D2 | Off | 55.3 | 295.0 | 406 | 152 | 0.1/0.6 | 1.86 | N.A. | 148 | 1.02±0.01 |
| D3 | Off | 80.1 | 294.5 | 278 | 147 | 0.9/1.6 | 0.29 | N.A. | 147 | 1.59±0.02 |
| L1A | On | 19.3 | 296.3 | N.A. | 92.2 | 0.4/1.5 | 0.36 | 0.49 | 120 | 1.66±0.03 |
| L1B | On | 55.1 | 298.5 | N.A. | 94.8 | 0.6/1.1 | 0.90 | 0.49 | 127 | 3.11±0.03 |
| L1C | On | 80.4 | 300.5 | N.A. | 88.4 | 0.3/1.1 | 0.71 | 0.49 | 118 | 3.35±0.02 |
| L2 | On | 20.4 | 297.0 | 123 | 87.8 | 0.3/1.7 | 0.30 | 0.51 | 120 | 1.66±0.04 |
| L3 | On | 55.2 | 299.3 | 120 | 82.3 | 0.2/1.9 | 1.79 | 0.86 | 120 | 2.54±0.21 |
| L4 | On | 80.7 | 298.7 | 131 | 78.0 | 0.2/0.4 | 0.28 | 1.54 | 120 | 5.22±0.19 |
| L5 | On | 21.0 | 296.9 | 130 | 78.1 | 0.1/1.35 | 64.8 | 0.15 | 60 | 1.70±0.14 |
| D4 | Off | 21.3 | 297.5 | 122 | 96.8 | <0.1/1.9 | 64.4 | 0.58 | 60 | 0.34±0.01 |
| L6 | On | 20.8 | 297.6 | N.A. | 90.3 | 68.7/109.5 | 1.52 | 0.24 | 132 | 1.67±0.07 |
| L7 | On | 21.3 | 296.9 | 121 | 76.8 | 77.0/110.7 | 5.79 | 0.26 | 207 | 2.78±0.12 |

[a] The data was obtained and organized from laboratory data reported by Park and Jang (2016). "D" denotes experiments under dark condition. "L" denotes experiments with UV light.
5   [b] The accuracy of RH is ±5%. The accuracy of temperature is ±0.5 K.
    [c] The mass concentration of ATD particles were calculated combining SMPS data, OPC data, the density of dust particles (2.65 g cm⁻³), and the particle size distribution (<3μm). The ppb s associated with the dust particle mass concentration were ±6%.
    [d] The errors associated with the observation of SO₂, NO, NO₂, O₃ and NH₄⁺ were ±0.9%, ±12.5%, ±6.9%, ±0.2% and ±5.0%, respectively.
    [e] The duration is the simulation time from the beginning of the experiment to the end of the experiment.
10  [f] Sulfate concentrations were measured at the end of experiments using PILS–IC. The measurements were not corrected for the particle loss rate to the wall, but corrected for the indigenous sulfate from dust particles.





**Table 2. Outdoor chamber experiment condition for SO₂ heterogeneously photooxidation on the ATD particles at variety initial concentration of SO₂, dust particle and NO₂.**

| Exp. Date | Purpose | RH[a] (%) | Temp.[a] (K) | simulation Time (EST) | Initial Concentration[b] | | | | |
|---|---|---|---|---|---|---|---|---|---|
| | | | | | ATD dust[c] ($\mu g\ m^{-3}$) | $SO_2$ (ppb) | $NO/NO_2$ (ppb) | $O_3$ (ppb) | $NH_4^+$ ($\mu g\ m^{-3}$) |
| 28/3/2015 | SO₂ | 18–67 | 277.1–301.9 | 11:10–16:30 | N.A. | 60.1 | 0.1/0.9 | 6.3 | 0.96 |
| | SO₂ & dust | 24–71 | 277.8–301.5 | 10:50–16:30 | 290.1 | 56.4 | 0.1/0.7 | 0.7 | 0.93 |
| 16/6/2015 | Low dust | 15–49 | 286.7–313.0 | 8:40–15:30 | 90.1 | 100.0 | 0.1/0.7 | 0.7 | 0.93 |
| | High dust | 16–48 | 287.0–311.5 | 9:30–15:30 | 403.7 | 120.1 | 1.1/1.0 | 5 | 0.59 |
| 12/11/2015 | Low SO₂ | 24–71 | 277.8–301.5 | 8:40–17:30 | 239.2 | 119.0 | 0.5/2.0 | 3.0 | 0.30 |
| | High SO₂ | 14–42 | 296.2–325.0 | 9:00–17:30 | 229.0 | 271.6 | 0.2/2.1 | 2.6 | 0.88 |
| 26/11/2015 | Low NO$_x$ | 19–49 | 288.1–308.6 | 9:30–15:30 | 229.6 | 89.9 | 19/3.5 | 10.0 | 0.74 |
| | High NO$_x$ | 20–45 | 288.5–309.1 | 9:00–15:30 | 322.3 | 115.3 | 70/127.7 | 3.1 | 0.61 |
| 05/11/2016 | Low NO$_x$ | 26–92 | 287.0–309.6 | 9:40–16:15 | 205.5 | 39.1 | 26.4/13.1 | 5.9 | 0.47 |
| | High NO$_x$ | 35–97 | 287.1–309.4 | 10:00–16:15 | 166.6 | 45.6 | 49.8/47.8 | 7.2 | 0.20 |
| 22/11/2016 | Low NO$_x$ | 19–86 | 275.0–306.5 | 7:00–15:30 | 150.3 | 44.5 | 14.4/8.7 | 4.2–91[d] | 0.42 |
| | High NO$_x$ | 26–89 | 275.2–306.7 | 7:30–15:30 | 118.6 | 44.0 | 30.2/38.5 | 5.1–70[d] | 0.38 |

[a] The accuracy of RH is ±5%. The accuracy of temperature is ±0.5 K.

[b] The errors associated with the observation of SO₂, NO, NO₂, O₃, NH₄⁺ and the concentration of dust particle mass were ±0.9%, ±12.5%, ±6.9%, ±0.2%, 5.0±%
5 and ±6%, respectively.

[c] The mass concentration of ATD particles were calculated combining SMPS data, OPC data, the density of dust particles (2.65 g cm⁻³), and the particle size distribution (<3μm) .

[d] 30 ppb of isoprene was injected into both chambers to increase the concentration of O₃.





**Table 3. Dust–phase heterogeneous reactions and their rate constants in the presence of ATD particles.**

| | Reaction[a] | Rate constant[b] | $k_1$ | $k_2$ | $k_3$ | Reference[b] | Note[c] |
|---|---|---|---|---|---|---|---|
| | *Partitioning* | | | | | | |
| 1 | $SO_2 + Dust \rightarrow SO_2(d) + Dust$ | d | $1\times10^{-8}$ | | | AR05, HZ15 | R7 |
| 2 | $SO_2(d) \rightarrow SO_2$ | e | $2\times10^9$ | 3100 | 0.013 | AR05, HZ15 | R8 |
| 3 | $O_3 + Dust \rightarrow O_3(d) + Dust$ | d | $1\times10^{-8}$ | | | MU03, US01 | |
| 4 | $O_3(d) \rightarrow O_3$ | f | $1\times10^{10}$ | 2700 | | MU03, US01 | |
| 5 | $NO_2 + Dust \rightarrow NO_2(d) + Dust$ | d | $1\times10^{-8}$ | | | CW84 | |
| 6 | $NO_2(d) \rightarrow NO_2$ | f | $5\times10^8$ | 2500 | | CW84 | |
| 7 | $HNO_3 + Dust \rightarrow HNO_3(d) + Dust$ | d | $1\times10^{-8}$ | | | SW81, Sc84 | |
| 8 | $HNO_3(d) \rightarrow HNO_3$ | e | $1\times10^{15}$ | 8700 | 15.4 | SW81, Sc84 | |
| 9 | $HONO + Dust \rightarrow HONO(d) + Dust$ | d | $1\times10^{-8}$ | | | BK96 | |
| 10 | $HONO(d) \rightarrow HONO$ | e | $1\times10^{10}$ | 4900 | 0 | BK96 | |
| 11 | $N_2O_5 + Dust \rightarrow HNO_3(d) + Dust$ | d | $7.3\times10^{-3}$ | | | WS09 | R20 |
| | *Dust phase* | | | | | | |
| 1 | $Dust + h\nu \rightarrow Dust + e\_h$ | g | $j_{[ATD]}$ | | | estimated | R10 |
| 2 | $e\_h \rightarrow energy$ | h | $1\times10^{-2}$ | | | estimated | R11 |
| 3 | $e\_h + O_2 \rightarrow OH(d)$ | i | $1\times10^{-22}$ | 2.3 | | estimated | R12 |
| 4 | $SO_2(d) \rightarrow SO_4^{2-}(d)$ | h | $5\times10^{-6}$ | | | estimated | R9 |
| 5 | $SO_2(d) + OH(d) \rightarrow SO_4^{2-}(d)$ | h | $1\times10^{-12}$ | | | estimated | R13 |
| 6 | $SO_2(d) + O_3(d) \rightarrow SO_4^{2-}(d) + O_2$ | h | $2\times10^{-11}$ | | | estimated | R14 |
| 7 | $e\_h + O_3(d) \rightarrow OH(d) + O_2$ | h | $1\times10^{-12}$ | | | estimated | R15 |
| 8 | $NO_2(d) \rightarrow NO_3^-(d)$ | h | $6\times10^{-5}$ | | | estimated | R18 |
| 9 | $e\_h + NO_2(d) \rightarrow HONO(d)$ | h | $6\times10^{-12}$ | | | estimated | R16 |
| 10 | $HONO(d) + h\nu \rightarrow OH(d) + NO$ | g | $j_{[HONO\_to\_OH]}$ | | | | R17 |
| 11 | $NO_2(d) + OH(d) \rightarrow NO_3^-(d)$ | h | $1\times10^{-10}$ | | | estimated | R19 |
| 12 | $NO_3^-(d) + Salt(d) \rightarrow NO_3^-(d\_salt)$ | h | $1\times10^{-8}$ | | | estimated | |
| 13 | $SO_4^{2-}(d) + Salt(d) \rightarrow SO_4^{2-}(d\_salt)$ | h | $1\times10^{-8}$ | | | estimated | |
| 14 | $NO_3^-(d\_salt) + SO_4^{2-}(d) \rightarrow SO_4^{2-}(d\_salt)$ | h | $4\times10^{-13}$ | | | estimated | |

[a] The unit of the chemical species (except dust) is molecule cm[-3] for both partitioning process and the dust phase chemistry. The unit of the dust for model input is mass concentration ($\mu$g m[-3]) and is multiplied by a factor of 2.45×10[10] for simulation.

[b] The rate constant parameters, which are noted as "estimated", are determined using the simulation of indoor chamber data(Park and Jang, 2016) (see Sect. 3). AR05, Adams et al. (2005); BK96, Becker et al. (1996); CW84, Chameides (1984); HZ15, Huang et al. (2015); MU03, Michel et al. (2003); Sc84, Schwartz (1984); SW81, Schwartz and White (1981); US01, Underwood et al. (2001); WS09, Wagner et al. (2009). The unit of reaction rate constants is s[-1] for first–order reactions, cm[3] molecule[-1] s[-1] for second–order reactions (except adsorption reactions).

[c] The reactions are noted with the numbers associated with the reaction in main text.

[d] Rate constant $k = k_1\,\omega\,f_{dust,M2S}\,/\,4$, where $\omega = \sqrt{8\,R\,T/(\pi\,MW)}$ (m s[-1]) and $f_{dust,M2S} = 3.066 \times 10^{-6}$ (m[2] $\mu$g). R is the ideal gas constant and MW (g mol[-1]) is the molecule weight of chemical species. The unit of the rate constants for the sorption reactions is m[3] m[-2] s[-1].

[e] Rate constant $k = k_1 \exp\left(-\frac{k_2}{T}\right)/(F_{water}(1 + k_3/[H^+]))$, where $F_{water} = \exp(4.4RH) + 3.7\exp(4.4RH)\frac{[NO_3^-]}{[Dust]} + \frac{M_{in,water}}{[Dust]}$. [H$^+$] and $M_{in,water}$ are dynamically calculated based on thermodynamic model (E–AIM II) (Clegg et al., 1998; Wexler and Clegg, 2002; Clegg and Wexler, 2011).

[f] Rate constant $k = k_1 \exp\left(-\frac{k_2}{T}\right)$.





$^{g}$ Photocatalytic reaction. The cross sections and quantum yields of dust are estimated (see Sect. 2.2). The cross sections and quantum yields of HONO(d) are taken from Bongartz et al. (1991) and Atkinson et al. (1997), respectively.

$^{h}$ Rate constant $k = k_1$.

5   $^{i}$ Rate constant $k = k_1 \exp(k_2)$.



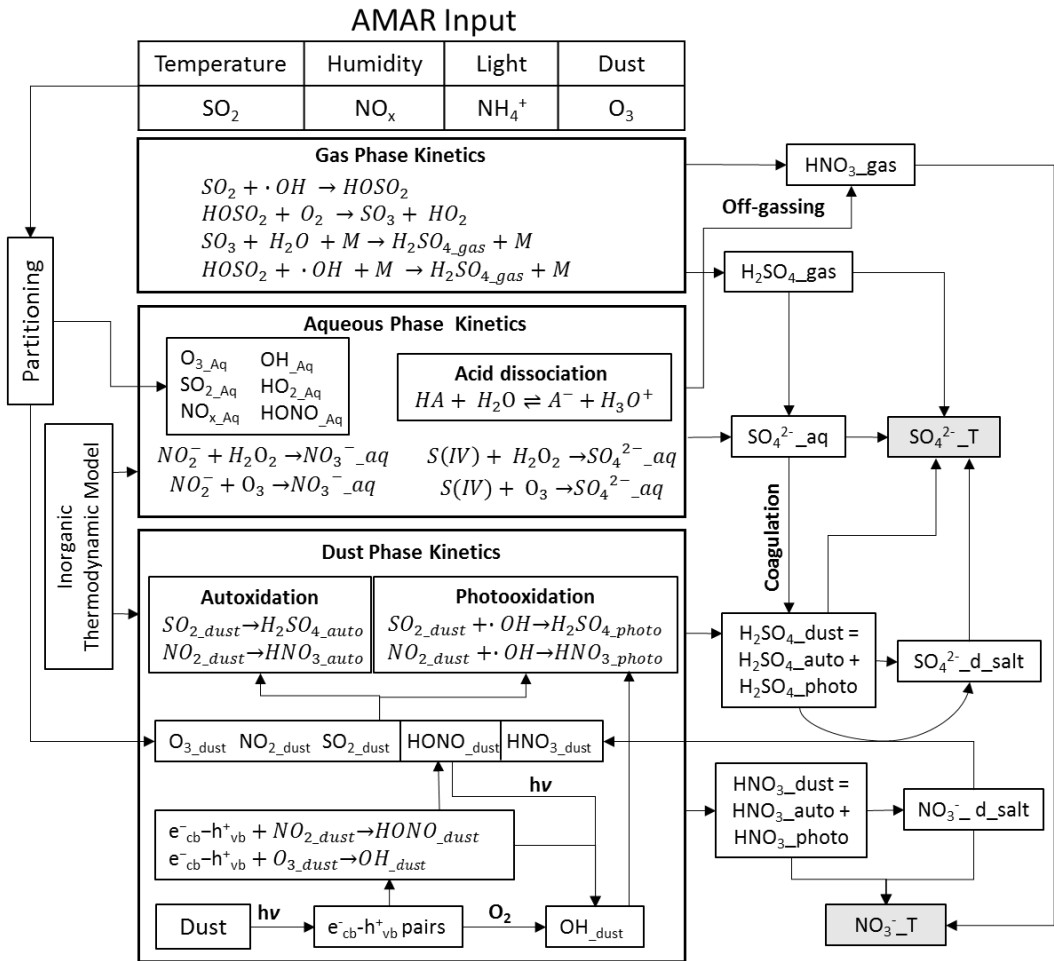

Figure 1. The overall schematic of the AMAR model to simulate heterogeneous $SO_2$ oxidation. For the description of chemical species, gas phase, aqueous phase and dust phase are symbolized as "gas", "aq" and "dust", respectively. $SO_4^{2-}$_T, $H_2SO_4$_gas, $SO_4^{2-}$_aq and $H_2SO_4$_dust are the total sulfate formation and the formation of sulfate from gas phase, aqueous phase and dust phase, respectively. $SO_4^{2-}$_d_salt and $NO_3^-$_d_salt are the neutralized sulfate and nitrate in dust phase.





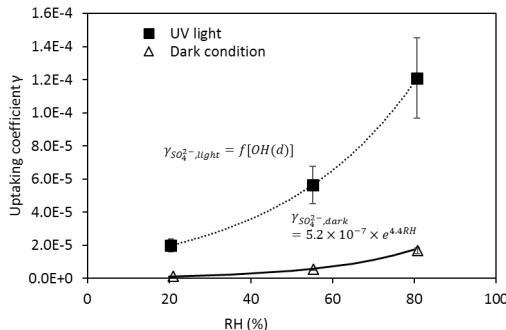

Figure 2. Uptake coefficient (γ) of SO$_2$ in the presence of the ATD particles under dark condition and UV light condition. The values of γ were obtained by kinetic model using indoor
5   experimental data. The $\gamma_{SO_4^{2-},light}$ is correlated to concentration of OH radicals and RH (%). The $\gamma_{SO_4^{2-},dark}$ is a function of RH. The error bar of γ was derived from the model uncertainty.





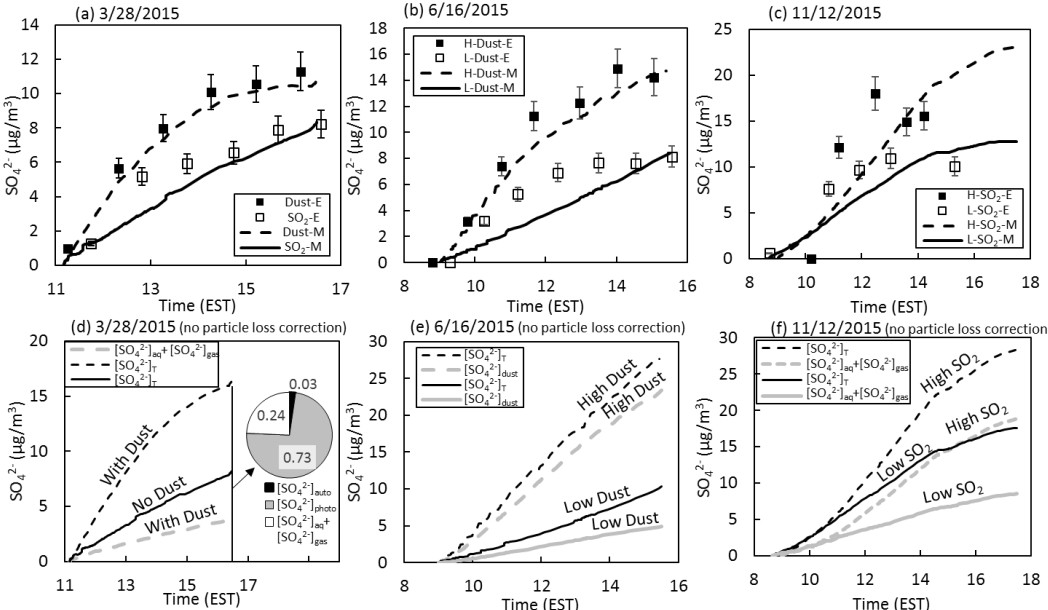

Figure 3. Time profiles of total sulfate concentration ($SO_4^{2-}$, μg m$^{-3}$) in the UF–APHOR. "E" denotes the experimentally observed sulfate ($[SO_4^{2-}]_T$) and "M" denotes the model–predicted sulfate. Sulfate concentrations were measured using PILS–IC during the experiments. The "H" and "L" represent the high and the low initial concentrations of chemical species. The errors associated with the concentration of sulfate is ±10% originated form the PILS–IC measurement. (a) Sulfate formation with and without ATD particles ($SO_2$ 60 ppb vs. $SO_2$ 56 ppb and dust 290 μg m$^{-3}$). (b) The high and low loadings of dust particles (dust 90 μg m$^{-3}$ and $SO_2$ 100 ppb vs. dust 404 μg m$^{-3}$ and $SO_2$ 120 ppb). (c) The high and the low concentrations of $SO_2$ ($SO_2$ 119 ppb and dust 239 μg m$^{-3}$ vs. $SO_2$ 272 ppb and dust 230 μg m$^{-3}$). For Fig. 3(a), 3(b) and 3(c), the simulations included the chamber dilution and the wall process of gaseous compounds and particles (Sect. S1). For Fig. 3(d), 3(e) and 3(f), the wall process for the particle loss was excluded to estimate the influence of ATD particles on sulfate formation without the chamber artefacts. In Fig. 3(d), 3(e) and 3(f), total sulfate was decoupled into the sulfate originated from dust chemistry and that originating from non–dust chemistry ($[SO_4^{2-}]_a + [SO_4^{2-}]_{gas}$). The pie chart inserted into Fig. 3(d) illustrates how total sulfate is attributed to major pathways at the end of the experiment.





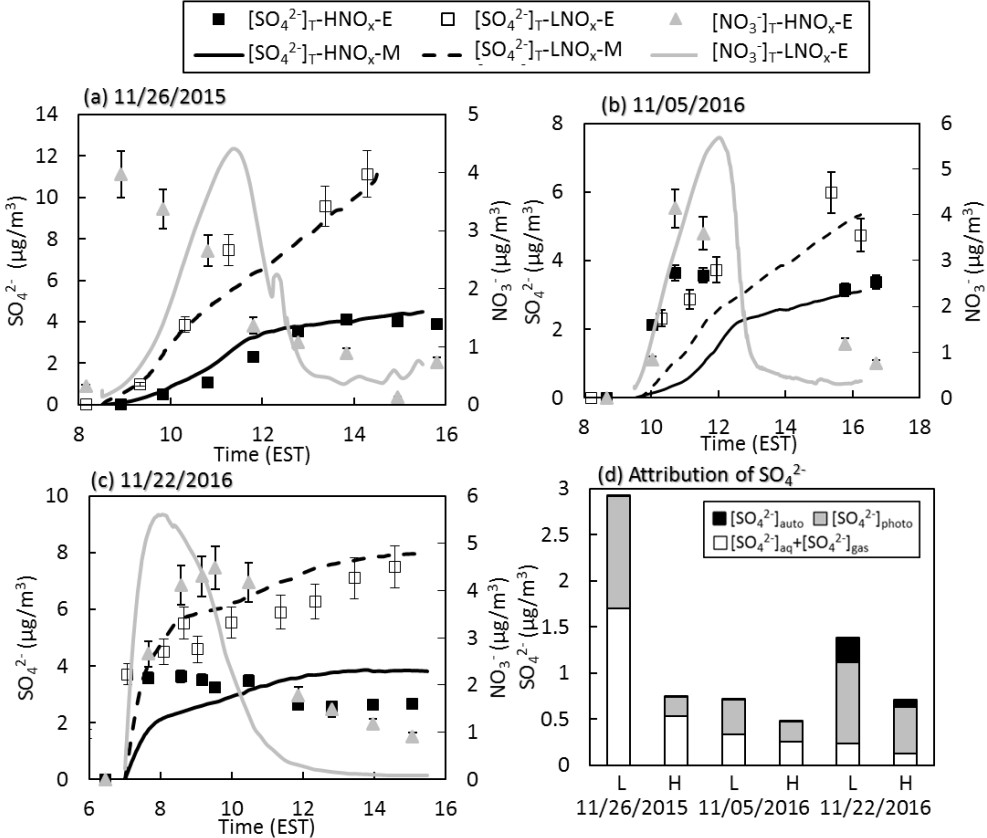

Figure 4. Time profiles of total sulfate concentration ($[SO_4^{2-}]_T$, µg/m$^3$) and nitrate concentration ($[NO_3^-]_T$, µg m$^{-3}$) in the dual chamber experiments using UF–APHOR at the two different $NO_x$

5   levels. The concentrations of sulfate and nitrate were measured using PILS–IC during the experiments. The error bars of the concentration of sulfate and nitrate is ±10% originated form the PILS–IC measurement. The detailed experimental conditions of Fig. 4(a) ($SO_2$ 90 ppb, $NO_x$ 23 ppb and dust 230 µg m$^{-3}$ vs. $SO_2$ 115 ppb, $NO_x$ 198 ppb and dust 322 µg m$^{-3}$), Fig. 4(b) ($SO_2$ 39 ppb, $NO_x$ 40 ppb and dust 206 µg m$^{-3}$ vs. $SO_2$ 46 ppb, $NO_x$ 98 ppb and dust 167 µg m$^{-3}$), and

10  Fig. 4(c) ($SO_2$ 45 ppb, $NO_x$ 23 ppb and dust 150 µg m$^{-3}$ vs. $SO_2$ 44 ppb, $NO_x$ 69 ppb and dust 119 µg m$^{-3}$) are shown in Table 2. Figure 4(d) shows how total sulfate is attributed to aqueous phase reaction (sulfate formation in gas phase + sulfate formation in inorganic salted inorganic aqueous phase) ($[SO_4^{2-}]_{aq}+[SO_4^{2-}]_{gas}$), dust–phase autoxidation ($[SO_4^{2-}]_{auto}$), and dust photochemistry ($[SO_4^{2-}]_{photo}$) at the end of the experiments. For Fig. 4(c), 30 ppb of isoprene was

15  injected to both chambers to yield a high concentration of ozone (90 ppb). "E" denotes the experimental observation and "M" denotes the simulation using the AMAR module. "H" and "L" represent the high and low initial concentration of $NO_x$ within the one set of dual chamber experiment. The chamber dilution and the wall process of gaseous compounds and particles were included in the simulation (Sect. S1).



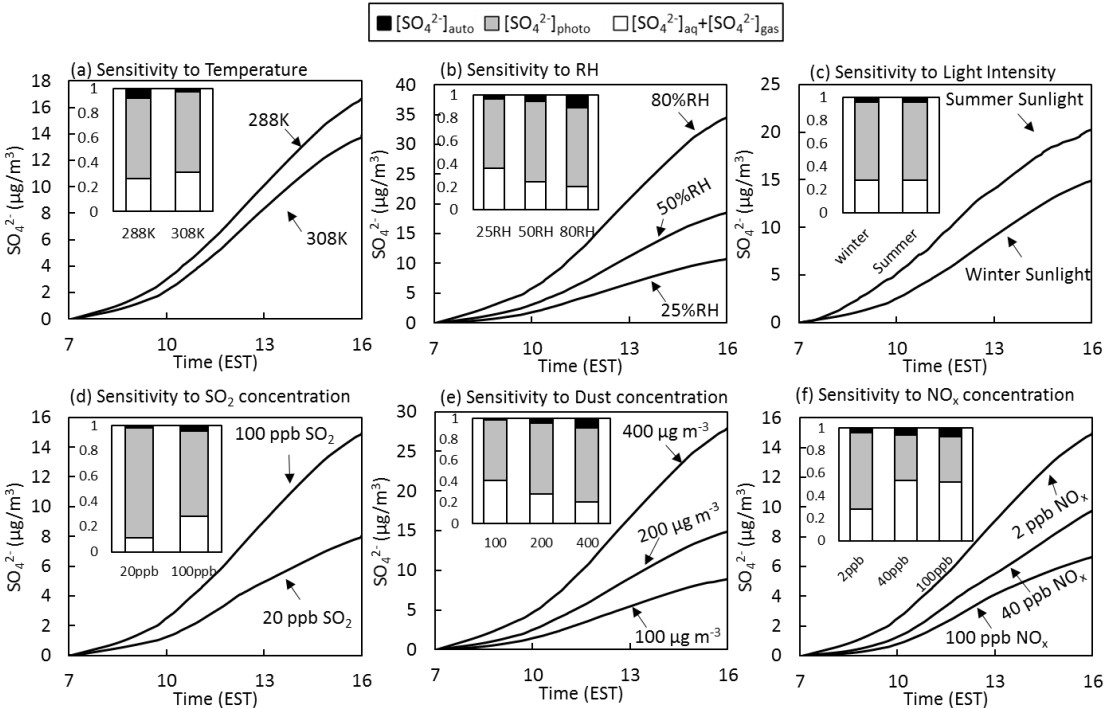

Figure 5. Sensitivity test of AMAR model to (a) temperature at 288K and 308K; (b) RH at 25%,
50% and 80%; (c) sunlight profiles of summertime (11 May 2015) and wintertime (22 November
2016) at Gainesville, Florida (latitude/longitude: 29.64185°/–82.347883°); (d) the concentration
of $SO_2$; (e) the concentration of dust particles; and (f) the $NO_x$ concentration. The stacked column
chart in each figure illustrates how total sulfate is attributed to major pathways at the end of each
experiment. For the sensitivity test, the chamber simulation is conducted with 100 ppb of initial
$SO_2$, 2 ppb of initial $NO_2$, 2 ppb of initial $O_3$ and 200 μg m$^{-3}$ of ATD particles at T = 298K and
RH = 40% under ambient sunlight on 22 November 2016. $NO_x$ (rate of flux = 2.7×10$^6$, s$^{-1}$) and
isoprene (rate of flux = 2.7×10$^6$, s$^{-1}$) were constantly added to simulate chamber dilution. The
simulation was performed without considering the particle loss to the chamber wall.