# Peer review of "Modelling Atmospheric Mineral Aerosol Chemistry to Predict Heterogeneous Photooxidation of SO2"

_Atmospheric Chemistry and Physics, 2017_

## Referee Comment (RC1) · Anonymous Referee #1 · 30 Mar 2017

This paper presents an interesting contribution aiming at the development of new numerical model of dust (photo)chemistry. It describes in some detail the different key process that are implemented. Clearly, the use of surface photochemistry is a novel and valuable contribution. However, a few aspects of this modelling studies are raising a few important questions.

The partitioning between the gas and condensed phase are treated in a similar way, despite being fundamentally different in nature. For a solid surface, the adsorption and desorption processes do follow a different formalism, typically through a Langmuir-Hinshelwood formalism which takes into account a given number of adsorption sites. The consequences is that adsorption decreases with time or increasing concentration,

while here it is simulated in a constant way with time. . . how can you justify such an assumption? Also products such as sulfate are probably staying on the surface, hereby also using adsorption sites i.e., poisoning the surface. How would your model change if you implement such time/concentration dependence?

Why having chosen to simulate deliquesced sea-salt and dust in the same code? What is the link between both objects/themes? Can you justify such a choice? Also the text mentions sea-salt and the partitioning process is described for aqueous sulfuric acid particles. Please harmonize the different part of your manuscript.

The characteristic time for adsorption is very different between aqueous and dust particles. Is this physically justified, bearing in mind that those processes are mostly related to gas phase and aerosols properties.

Too many rate constant are estimated without any justification. Please justify and explain your estimations.

The agreement with the chamber data has to be described in a more quantitative way. By looking at the figures, one may have the impression that the agreement is not as good as described in the text.

Without these changes, I do not think that this manuscript is suitable for publication in ACP.

———————————————

---

## Author Comment (AC1) · 23 May 2017

**Response to Reviewers' comments (Manuscript Ref. NO.: acp-2017-120)**
We would like to thank the reviewer for the thoughtful comments. We have carefully studied these comments and made the corrections. The quality of this manuscript has been greatly improved due to the valuable suggestions.

**Reviewer #1:**

**Overall Comment**
This paper presents an interesting contribution aiming at the development of new numerical model of dust (photo)chemistry. It describes in some detail the different key process that are implemented. Clearly, the use of surface photochemistry is a novel and valuable contribution. However, a few aspects of this modelling studies are raising a few important questions.

**Comment 1:** The partitioning between the gas and condensed phase are treated in a similar way, despite being fundamentally different in nature. For a solid surface, the adsorption and desorption processes do follow a different formalism, typically through a Langmuir-Hinshelwood formalism which takes into account a given number of adsorption sites. The consequences is that adsorption decreases with time or increasing concentration, while here it is simulated in a constant way with time. how can you justify such an assumption? Also products such as sulfate are probably staying on the surface, thereby also using adsorption sites i.e., poisoning the surface. How would your model change if you implement such time/concentration dependence?

**Response:** We assumed that the gas-particle partitioning onto dust is operated by an absorption process (Eq. 7) by several reasons (see section 3.2.1). First, unlike pure metal oxide which is governed by the adsorptive partitioning, the composition of authentic mineral dust such as Arizona Test Dust (ATD) is complex. The fresh ATD contains inorganic salts that are hygroscopic and form the water film above efflorescence relative humidity or deliquescence relative humidity. Second, the partitioning process is dynamic due to the formation of various hygroscopic salts of sulfate and nitrate due to the reaction of alkaline carbonates and metal oxides with inorganic acids (sulfuric acid and nitric acid). Third, the sulfate formation in our study increased as increasing humidity due to the dissolution of tracers into the water layer (see section 3.2.1). If partitioning is processed by the adsorptive mode, water molecules compete for the site with tracers and reduce partitioning of tracers (Cwiertny et al., 2008). The amount of the surface water on dust particles, which was measured using FTIR (submitted in the other journal), was multi-layered.

**Comment 2:** Why having chosen to simulate deliquesced sea-salt and dust in the same code? What is the link between both objects/themes? Can you justify such a choice? Also the text mentions sea-salt and the partitioning process is described for aqueous sulfuric acid particles. Please harmonize the different part of your manuscript.

**Response:** We have never mentioned about the deliquesced sea-salt in this study. For the formation of sulfuric acid in inorganic salted aqueous aerosol ($SO_4^{2-}$-$NH_4^+$-$H_2O$ system) (Section 3.1.3), we employed the aqueous phase kinetic reactions previously reported in literatures (Lee, 1984; Strehlow and Wagner, 1982; Gratzel et al., 1970; Graedel and Weschler, 1981; Treinin and Hayon, 1970; Lee and Lind, 1986; Damschen and Martin, 1983; Liang and Jacobson, 1999; Hoyle et al., 2016).

**Comment 3:** The characteristic time for adsorption is very different between aqueous and dust particles. Is this physically justified, bearing in mind that those processes are mostly related to gas phase and aerosols properties.

**Response:** The characteristic time of the uptake process of gas into aqueous phase or dust phase is calculated for gas-phase diffusion, liquid phase diffusion, establishing equilibrium at the interface and the reactions in gas, aqueous, and dust phases (Finlayson-Pitts and Pitts Jr, 1999). Table S2 has been added to supporting information in revised manuscript.

**Table S2. The calculation of the characteristic time of the major processes and reactions**

| Type of process | Characteristic time | Aqueous phase system | Dust system |
|---|---|---|---|
| $r$: the particle radius
$D_g$: diffusion coefficient in gas
$D_l$: diffusion coefficient in aqueous phase
$H$: Henry's constant
$\alpha$: the mass accommodation coefficient (0.5)
$u_{av}$ is the mean thermal speed. | | $r = 50$ nm
$k_{SO2,g}= 1\times10^{-12}$ s$^{-1}$ molecule$^{-1}$ cc
$[OH] = 1\times10^6$ molecules/cc
$k_{HSO3} = 3\times10^{-3}$ s$^{-1}$ molecule$^{-1}$ cc
$[OH]_{aq} = 0.1$ molecules/cc | $r = 350$ nm (average)
$r_{dust\_aq}$: the average thickness of the water layer on dust particles (40 nm)
$k_{SO2,g}= 1\times10^{-12}$ s$^{-1}$ molecule$^{-1}$ cc
$[OH] = 1\times10^6$ molecules/cc
$k_{SO2,dust} = 1\times10^{-12}$ s$^{-1}$ molecule$^{-1}$ cc
$[OH]_{aq}=2\times10^9$ molecules/cc |
| Gas diffusion | $\dfrac{r^2}{\pi^2 D_g}$ | $2.4\times10^{-11}$ s | $2.1\times10^{-10}$ s |
| Diffusion in aqueous phase | $\dfrac{r_{aq}^2}{\pi^2 D_l}$ | $1.9\times10^{-7}$ s | |
| Diffusion in the water layer on dust | $\dfrac{r_{dust\_aq}^2}{\pi^2 D_l}$ | | $1.2\times10^{-7}$ s |
| Equilibrium between gas and particle | $D_l\left(\dfrac{4HRT}{\alpha u_{av}}\right)^2$ | $7.1\times10^{-10}$ s | $7.1\times10^{-10}$ s |
| Reaction in gas phase | $\dfrac{1}{[OH]k_{SO2,g}}$ | $1\times10^6$ s | $1\times10^6$ s |
| Reaction in aqueous phase | $\dfrac{1}{[OH]_{aq}k_{HSO3,aq}}$ | $2\times10^3$ s | |
| Reaction in dust phase | $\dfrac{1}{[OH]_{dust}k_{SO2,dust}}$ | | $5\times10^2$ s |

For both the aqueous system and the dust system, the characteristic time of all reactions in gas and particles are much greater than diffusion in gas or particle phases and equilibrium processes (partitioning and dissociation of acids). Thus, the reactions of chemical species are not affected by the time reached to equilibrium or diffusion processes. In the model, both absorption and desorption rates of chemical species were set to much faster than their reaction rates in all three phase (last paragraph in Section 3.1.2). Furthermore, the time of diffusion in the liquid phase is longer than both gas diffusion and the time for reaching to equilibrium as shown in Table S2.

**Comment 4:** Too many rate constants are estimated without any justification. Please justify and explain your estimations.

**Response:** Most of the rate constants shown in Table 3 were estimated using the indoor chamber data obtained in the previous study (Park and Jang, 2016). The rate constants of R10 (electron-hole production) and R11 (recombination of electron-hole) in the manuscript is estimated using Eq. 10 (photoactivation rate, $J_{ATD}$) in the manuscript (Section 3.2.3). The rate constant of R13 (reaction of $SO_2$ with dust-phase OH radicals) is set to the same reaction rate constant for the reaction of $SO_2$ with OH radicals in gas phase. Without sunlight, autoxidation of $SO_2$ (R9) is dominant in dust phase and its rate constant was obtained from indoor chamber data under various humidity conditions (Exp. D1-D3 in Table 1). With sunlight, the photochemical reaction is the major source for sulfate production. Using the same approach with autoxidation, the rate constant of R12 was estimated under different humidity conditions. Also, the rate constants of R14 (heterogeneous autoxidation of $SO_2$ in the presence of ozone) and R15 (heterogeneous oxidation of $O_3$) were estimated using experiments D4 and L5 in Table 1, respectively. The rate constants of R18 (heterogeneous autoxidation of $NO_2$) and R19 (heterogeneous photocatalytic oxidation of $NO_2$) were estimated using experiments D5 and L7 in Table 1, respectively.

**Comment 5:** The agreement with the chamber data has to be described in a more quantitative way. By looking at the figures, one may have the impression that the agreement is not as good as described in the text.

**Response:** We agree with the reviewer. After carefully searching the errors, we found that mistakes in the estimation of the aerosol water content and aerosol acidity due to the incorrect input parameter for the titration of sulfuric acid with ammonia. The FS value, a numeric number to dynamically represent the composition of the sulfate-ammonium aerosol system, ranges from 0.34 (ammonium sulfate) to 1.0 (sulfuric acid). In the previous simulation, FS was computed sometimes at out of range due to the incorrect input of ammonia data. By the correction of this error, the model prediction in Fig. 3 were improved. In addition to the correction of input errors, we found some contamination in the $NO_2$ tank by nitric acid for the experiments with $NO_2$ (11/26/2015, 11/05/2016 and 11/22/2016). We conducted additional experiments with the new $NO_x$ tank (Table 2 in the revised manuscript) and applied these new data to model evaluations (Figure 4 and Table 2 in the revised manuscript). The simulation of $SO_2$ oxidation in the presence of $NO_2$ was much improved.

[revised manuscript text omitted]

Hoyle, C. R., Fuchs, C., Jarvinen, E., Saathoff, H., Dias, A., El Haddad, I., Gysel, M., Coburn, S. C., Trostl, J., Bernhammer, A. K., Bianchi, F., Breitenlechner, M., Corbin, J. C., Craven, J., Donahue, N. M., Duplissy, J., Ehrhart, S., Frege, C., Gordon, H., Hoppel, N., Heinritzi, M., Kristensen, T. B., Molteni, U., Nichman, L., Pinterich, T., Prevot, A. S. H., Simon, M., Slowik, J. G., Steiner, G., Tome, A., Vogel, A. L., Volkamer, R., Wagner, A. C., Wagner, R., Wexler, A. S., Williamson, C., Winkler, P. M., Yan, C., Amorim, A., Dommen, J., Curtius, J., Gallagher, M. W., Flagan, R. C., Hansel, A., Kirkby, J., Kulmala, M., Mohler, O., Stratmann, F., Worsnop, D. R., and Baltensperger, U.: Aqueous phase oxidation of sulphur dioxide by ozone in cloud droplets, Atmos Chem Phys, 16, 1693-1712, 10.5194/acp-16-1693-2016, 2016.

Krueger, B. J., Grassian, V. H., Laskin, A., and Cowin, J. P.: The transformation of solid atmospheric particles into liquid droplets through heterogeneous chemistry: Laboratory insights into the processing of calcium containing mineral dust aerosol in the troposphere, Geophys Res Lett, 30, Artn 1148 10.1029/2002gl016563, 2003.

Lee, Y.-N.: Atmospheric aqueous-phase reactions of nitrogen species, Brookhaven National Lab., Upton, NY (USA), 1984.

Lee, Y. N., and Lind, J. A.: Kinetics of aqueous-phase oxidation of nitrogen (III) by hydrogen peroxide, Journal of Geophysical Research: Atmospheres, 91, 2793-2800, 1986.

Liang, J. Y., and Jacobson, M. Z.: A study of sulfur dioxide oxidation pathways over a range of liquid water contents, pH values, and temperatures, J Geophys Res-Atmos, 104, 13749-13769, Doi 10.1029/1999jd900097, 1999.

Liu, Y. J., Zhu, T., Zhao, D. F., and Zhang, Z. F.: Investigation of the hygroscopic properties of Ca(NO3)(2) and internally mixed Ca(NO3)(2)/CaCO3 particles by micro-Raman spectrometry, Atmos Chem Phys, 8, 7205-7215, 2008.

Park, J., and Jang, M.: Heterogeneous photooxidation of sulfur dioxide in the presence of airborne mineral dust particles, RSC Advances, 6, 58617-58627, 2016.

Strehlow, H., and Wagner, I.: Flash photolysis in aqueous nitrite solutions, Zeitschrift für Physikalische Chemie, 132, 151-160, 1982.

Treinin, A., and Hayon, E.: Absorption spectra and reaction kinetics of NO2, N2O3, and N2O4 in aqueous solution, J Am Chem Soc, 92, 5821-5828, 1970.

---

## Referee Comment (RC2) · Anonymous Referee #2 · 25 May 2017

This manuscript represents a model for evaluating the importance of dust in sulfate formation, particularly in adding the kinetics and mechanism of heterogeneous photo-catalytic reactions of SO$_2$ on mineral dust in the model. It is essential to consider the photooxidation of SO$_2$ in order to improve the accuracy of sulfate formation modeled in the atmosphere. Therefore, this study is of substantial interest. However, some major points should be carefully considered before it is published.

Major comments:

(1) The indoor chamber data shows that, in the absence of ATD particles, [SO$_4{}^{2-}$]$_T$ at 55% RH is two times larger than that at 19% RH (Table 1 L1A, B and C), but when RH increases to 80%, the enhancement of [SO$_4{}^{2-}$]$_T$ is not distinct. Additionally, in the

presence of ATD particles, $[SO_4^{2-}]_T$ is unexpectedly lower than that in the absence of ATD at 55% RH (Table 1 L3 and L1B), contrary to that at 80% RH (Table 1 L4 and L1C). However, these observations are not discussed in the manuscript and shown in the model.

(2) In addition to react with $SO_2$ and $NO_2$, OH radicals produced on the surface of particles under UV conditions can undergo heterogeneous reaction with particles as well as self-reactions, resulting in the significant decrease of OH radicals participate in the oxidation of $SO_2$ and $NO_2$, and subsequently overestimating sulfate and nitrate formation in the model. Furthermore, in addition to compete OH radicals with $SO_2$, the presence of $NO_2$ can also react with $SO_2$ on the surface of particles to promote sulfate formation at high RHs as like in aqueous phase. However, these mechanisms were not considered in dust phase in the model (Table S1).

(3) In Figure 3, it seems that modeled results are not in agreement with experimental observations at scenarios (a) without ATD particles and (b) low loadings of dust particles, especially for time-changing trends, meaning that the gas and aqueous phase reaction of $SO_2$ may be not well considered in the model. The authors should give explanations or speculations for this discrepancy in the manuscript.

(4) The authors estimated gas-particle partitioning constant of $NO_2$, $K_{d,NO2}$, based on the relationship between the Henry's law constants of $NO_2$ and $SO_2$ (Eq. 15), but $K_{d,O3}$ is obtained from literature results (Eq. 14). Is it reasonable to estimate $K_{d,NO2}$ according to Eq. (15)? And why $K_{d,NO2}$ and $K_{d,O3}$ are set based on different method since previous studies have investigated the heterogeneous reaction of $NO_2$ on mineral dust as well? Moreover, in Section 3.2.1 the authors considered the influence of RH on $K_{d,SO2}$, however, the expression of $K_{d,NO2}$ and $K_{d,O3}$, which is also closely related sulfate formation in the model, was not shown as a function of RH.

Minor comments:

Page 4 Line 27 "The detail description" should be "The detailed description"

Page 4 Line 21 The indoor chamber data of this study was obtained from our recent laboratory study (Park and Jang, 2016), however, $[SO_4^{2-}]_T$ values shown in Table 1 is different with Park and Jang (2016) reported. For example, Table 1 D1, L1 B and L8 in the manuscript corresponding to Table 1 D1, L1D and L8 in Park and Jang (2016), respectively.

Page 7 Line 19 and 21 "$SO_4^2$–$NH_4^+$–$H_2O$" should be "$SO_4^{2-}$–$NH_4^+$–$H_2O$".

Page 11 Line 17 Give more detailed description about $k_{auto}$o and $k_{OH,O2}$ derived from the indoor chamber data.

Page 14 Line 21 "L7 and L8 in Table 1" should be "L6 and L7 in Table 1".

---

## Author Comment (AC2) · 1 Jun 2017

**Response to Reviewers' comments RC2 (Manuscript Ref. NO.: acp-2017-120)**

We appreciate the referee for the time spent on our work and the constructive comments. The quality of our work has been improved greatly according to the thoughtful suggestions. The detailed responses to specific questions are presented in the following.

**Reviewer #2:**

**Overall Comment**

This manuscript represents a model for evaluating the importance of dust in sulfate formation, particularly in adding the kinetics and mechanism of heterogeneous photocatalytic reactions of $SO_2$ on mineral dust in the model. It is essential to consider the photooxidation of $SO_2$ in order to improve the accuracy of sulfate formation modeled in the atmosphere. Therefore, this study is of substantial interest. However, some major points should be carefully considered before it is published.

**Comment 1:** The indoor chamber data shows that, in the absence of ATD particles, $[SO_4^{2-}]_T$ at 55% RH is two times larger than that at 19% RH (Table 1 L1A, B and C), but when RH increases to 80%, the enhancement of $[SO_4^{2-}]_T$ is not distinct. Additionally, in the presence of ATD particles, $[SO_4^{2-}]_T$ is unexpectedly lower than that in the absence of ATD at 55% RH (Table 1 L3 and L1B), contrary to that at 80% RH (Table 1 L4 and L1C). However, these observations are not discussed in the manuscript and shown in the model.

**Response:** The data of Table 1 is obtained and reorganized from previous study (Park and Jang, 2016). In Table 1, $[SO_4^{2-}]_T$ is the observation of total sulfate concentration which is dependent of the initial $SO_2$, RH, dust concentration and the duration of the experiment. Figure 2 illustrates the impact of RH on uptake coefficient of $SO_2$ with and without UV light. The uptake coefficients which were determined using fitting the kinetic model to experimental data agree with those reported in the previous work by Park and Jang (2016). Exp. L1A, L1B and L1C in Table 1 were for $SO_2$ oxidation without dust particles and removed in the revised manuscript because they were not used for this paper.

**Comment 2:** In addition to react with $SO_2$ and $NO_2$, OH radicals produced on the surface of particles under UV conditions can undergo heterogeneous reaction with particles as well as self-reactions, resulting in the significant decrease of OH radicals participate in the oxidation of $SO_2$ and $NO_2$, and subsequently overestimating sulfate and nitrate formation in the model. Furthermore, in addition to compete OH radicals with $SO_2$, the presence of $NO_2$ can also react with $SO_2$ on the surface of particles to promote sulfate formation at high RHs as like in aqueous phase. However, these mechanisms were not considered in dust phase in the model (Table S1).

**Response:** In our model, the apparent rate constant of the formation of the dust-phase OH radicals is estimated using indoor chamber data. The synergistic effect of $NO_2$ on sulfate formation under

UV light is explained by the HONO production through the reaction of $NO_2$ with electrons or holes in dust phase (R16). HONO will then be decomposed via photolysis to form OH radicals (R17).

**Comment 3**: In Figure 3, it seems that modeled results are not in agreement with experimental observations at scenarios (a) without ATD particles and (b) low loadings of dust particles, especially for time-changing trends, meaning that the gas and aqueous phase reaction of $SO_2$ may be not well considered in the model. The authors should give explanations or speculations for this discrepancy in the manuscript.

**Response:** Please also fine the response to comment 5 from reviewer 1. We agree with the reviewer's comment. The estimation of aerosol water content was incorrect in the previous simulation due to the wrong input of aerosol acidity. By correcting this error, the model simulation of sulfate and nitrate has been greatly improved (Figs 3 and 4).

**Comment 4**: The authors estimated gas-particle partitioning constant of $NO_2$, $K_{d, NO2}$, based on the relationship between the Henry's law constants of $NO_2$ and $SO_2$ (Eq. 15), but $K_{d,O3}$ is obtained from literature results (Eq. 14). Is it reasonable to estimate $K_{d,NO2}$ according to Eq. (15)? And why $K_{d,NO2}$ and $K_{d,O3}$ are set based on different method since previous studies have investigated the heterogeneous reaction of $NO_2$ on mineral dust as well? Moreover, in Section 3.2.1 the authors considered the influence of RH on $K_{d,SO2}$, however, the expression of $K_{d,NO2}$ and $K_{d,O3}$. The concentration otration of which is also closely related sulfate formation in the model, was not shown as a function of RH.

**Response:** The estimation of the gas-dust partitioning coefficient of ozone is scaled using the gas-dust partitioning of $SO_2$ reported in literature and the ratio of Henry' constant of $SO_2$ to ozone, similar to $NO_2$ (the first sentence of Sect. 3.3.1). The partitioning process of tracers on the dust phase is treated as absorption and desorption process (please also see the response to the Comment 1 of Referee 1). The absorption process is influenced by the aerosol water content. Thus, we assume that relative ratio of the Henry's law constants normalized by constant of $SO_2$, is applicable to estimate the gas-dust partitioning coefficient of tracers. For example, the Henry's law constants of both $NO_2$ and ozone are $1.2 \times 10^{-2}$ mol $L^{-1}$ atm$^{-1}$ at 298K (Chameides, 1984) and they are 100 times smaller than that of $SO_2$ (1.2 mol $L^{-1}$ atm$^{-1}$)(Chameides, 1984). All gas-dust partitioning coefficients are dependent of humidity (see Eq. 7, 14, and 15).

**Minor Comments:**
**Comment 5:** Page 4 Line 27 "The detail description" should be "The detailed description"

**Response:** It was corrected.

**Comment 6:** Page 4 Line 21 The indoor chamber data of this study was obtained from our recent laboratory study (Park and Jang, 2016), however, $[SO_4^{2-}]_T$ values shown in Table 1 is different

with Park and Jang (2016) reported. For example, Table 1 D1, L1 B and L8 in the manuscript corresponding to Table 1 D1, L1D and L8 in Park and Jang (2016), respectively.

**Response:** The model simulation was performed against the data points over the course of the chamber experiment. The $[SO_4^{2-}]_T$ values in Table 1 are sourced from the last data point in time for each chamber experiment that previously reported by Park and Jang (2016). In order to drive the model parameters, the outliers were removed for some data sets. When data was available, different simulation time were used for some data sets. Data sets L1A, L1B, and L1C for $SO_2$ oxidation without dust particles were not used in this study. Thus, they were removed from Table 1. For the heterogeneous autoxidation of $SO_2$ in the presence of ozone, the newly produced indoor chamber data (Data D4 in Table 1) was used to drive the AMAR model. As described in the response to comment 5 of reviewer 1, we found the contamination of the $NO_2$ tank by nitric acid. Thus, we removed data sets L6 and L7 from Table 1. To insure the quality of the data used for the model, the new outdoor chamber data sets were produced for heterogeneous oxidation of $SO_2$ in the presence of $NO_x$ and they were applied to model development: data on 04/14/2017 in Table 2 for driving model parameters and two data sets on 04/25/2017 for model evaluation.

**Comment 7:** Page 7 Line 19 and 21 "$SO_4^2$–$NH_4^+$–$H_2O$" should be "$SO_4^{2-}$–$NH_4^+$–$H_2O$".
**Response:** This has been done.

**Comment 8:** Page 11 Line 17 Give more detailed description about $k_{autoo}$ and $k_{OH, O2}$ derived from the indoor chamber data.
**Response:** In the dark condition, the formation of sulfate is mainly from the autoxidation of $SO_2$. By fitting the predicted sulfate concentration to the experimental observation (D1- D3 in Table 1), the $SO_2$ autoxidation reaction rate constant ($k_{auto}$, $s^{-1}$) is semiempirically determined. Also see the last third sentence of Sect. 3.2.2. Using same approach, $k_{OH,O2}$ is first estimated using indoor chamber data (L1-L3 in Table 1) at RH 20%, 55% and 80% and then regressed against RH. Also see the last sentence of the second paragraph of Sect. 3.2.3.

**Comment 9:** Page 14 Line 21 "L7 and L8 in Table 1" should be "L6 and L7 in Table 1".
**Response:** It was corrected. Thank you very much for your comments.

---

## Editor Decision (ED1)

**1) Referee #1**
**Comment 1:** The partitioning between the gas and condensed phase are treated in a similar way, despite being fundamentally different in nature. For a solid surface, the adsorption and desorption processes do follow a different formalism, typically through a Langmuir-Hinshelwood formalism which takes into account a given number of adsorption sites. The consequences is that adsorption decreases with time or increasing concentration, while here it is simulated in a constant way with time. how can you justify such an assumption? Also products such as sulfate are probably staying on the surface, thereby also using adsorption sites i.e., poisoning the surface. How would your model change if you implement such time/concentration dependence?

**Response:** We assumed that the gas-particle partitioning onto dust is operated by an absorption process (Eq. 7) by several reasons (see section 3.2.1). First, unlike pure metal oxide which is governed by the adsorptive partitioning, the composition of authentic mineral dust such as Arizona Test Dust (ATD) is complex. The fresh ATD contains inorganic salts that are hygroscopic and form the water film above efflorescence relative humidity or deliquescence relative humidity. Second, the partitioning process is dynamic due to the formation of various hygroscopic salts of sulfate and nitrate due to the reaction of alkaline carbonates and metal oxides with inorganic acids (sulfuric acid and nitric acid). Third, the sulfate formation in our study increased as increasing humidity due to the dissolution of tracers into the water layer (see section 3.2.1). If partitioning is processed by the adsorptive mode, water molecules compete for the site with tracers and reduce partitioning of tracers (Cwiertny et al., 2008). The amount of the surface water on dust particles, which was measured using FTIR (submitted in the other journal), was multi-layered.

**Editor:** It seems to me that the only change in response to this comment was that you replaced all 'adsorption' by 'absorption' in the manuscript. However, this raises more questions than it adds to clarification. You argue that the hygroscopic fraction of the dust forms an aqueous phase where then the chemical processing in the 'dust phase' occurs. Thus, the extent of this processing will depend on the mass (volume) of the hygroscopic dust material. It is not clear to me why parameters such as the absorption rate constants (e.g. R5 and Equation 5) can be used that relate to a particle surface (m-2) and not to a (partial) particle volume.

This should be clarified throughout the manuscript. The text should be carefully checked for consistency (e.g. p. 15, l. 10 'adsorption-desorption') and conclusions should be refined (e.g. why is BET surface area needed (p. 20, l. 26) if the hygroscopic material determines the reactions medium?)

**2) Referee #1**
**Comment 4:** Too many rate constants are estimated without any justification. Please justify and explain your estimations.

**Response:** Most of the rate constants shown in Table 3 were estimated using the indoor chamber data obtained in the previous study (Park and Jang, 2016). The rate constants of R10 (electron-hole production) and R11 (recombination of electron-hole) in the manuscript is estimated using Eq. 10 (photoactivation rate, $J_{ATD}$) in the manuscript (Section 3.2.3). The rate constant of R13 (reaction of $SO_2$ with dust-phase OH radicals) is set to the same reaction rate constant for the reaction of $SO_2$ with

OH radicals in gas phase. Without sunlight, autoxidation of $SO_2$ (R9) is dominant in dust phase and its rate constant was obtained from indoor chamber data under various humidity conditions (Exp. D1-D3 in Table 1). With sunlight, the photochemical reaction is the major source for sulfate production. Using the same approach with autoxidation, the rate constant of R12 was estimated under different humidity conditions. Also, the rate constants of R14 (heterogeneous autoxidation of $SO_2$ in the presence of ozone) and R15 (heterogeneous oxidation of $O_3$) were estimated using experiments D4 and L5 in Table 1, respectively. The rate constants of R18 (heterogeneous autoxidation of $NO_2$) and R19 (heterogeneous photocatalytic oxidation of $NO_2$) were estimated using experiments D5 and L7 in Table 1, respectively.

**Editor:** In my opinion, your response is neither a justification nor an explanation why so many rate constants were estimated. Please add a more detailed discussion of uncertainties and background on these constants in order to fulfill the reviewer's inquiry.

Also, I noticed that e.g. $K_{d,SO2}$ is used for dry particles – how can this be justified?

**3) Referee #3:**

**Comment 2**: In addition to react with $SO_2$ and $NO_2$, OH radicals produced on the surface of particles under UV conditions can undergo heterogeneous reaction with particles as well as self-reactions, resulting in the significant decrease of OH radicals participate in the oxidation of $SO_2$ and $NO_2$, and subsequently overestimating sulfate and nitrate formation in the model. Furthermore, in addition to compete OH radicals with $SO_2$, the presence of $NO_2$ can also react with $SO_2$ on the surface of particles to promote sulfate formation at high RHs as like in aqueous phase. However, these mechanisms were not considered in dust phase in the model (Table S1).

**Response:** In our model, the apparent rate constant of the formation of the dust-phase OH radicals is estimated using indoor chamber data. The synergistic effect of $NO_2$ on sulfate formation under UV light is explained by the HONO production through the reaction of $NO_2$ with electrons or holes in dust phase (R16). HONO will then be decomposed via photolysis to form OH radicals (R17).

**Editor:** The reviewer's concern of missing reactions in your mechanism is well justified. How would the recombination of OH or their loss on particle surfaces affect your results? Please add a discussion about what it is known about such processes and to what extent they may compete with other processes in your system.

**4) Difference between dust chemistry and aqueous phase chemistry**

Editor: Related to the comment above, I do not understand the fundamental difference between the aqueous phase chemistry (Section 3.1.3) and dust chemistry (Section 3.2). Both are reactions that occur in a bulk aqueous phase and thus mechanistically they should be treated equally (even though different chemical reactions occur). Please justify the differentiation into two types of processes.

It should be stated throughout the manuscript that the 'dust phase' is also technically an aqueous phase

**5) Use of Henry's law constants**

a) What is the ionic strength and/or acidity of the aqueous phases? Is the application of Henry's law constants (for ideal solutions) justified? If not, how does this affect the results and the possibility of extrapolation of the derived rate constants to other conditions?

b) Is the pH sufficiently low that the uptake of SO2 can be indeed solely described by the physical Henry's law constant KH,SO2? This can be only applied if the solution is sufficiently acidic and no dissociation occurs; otherwise the effective Henry's law constant including dissociation should be included. Please justify.

**6) Language**

The whole manuscript should be carefully checked for proper use of English language. I list some rather unusual or unclear expressions below (line numbers refer to the marked-up manuscript that was attached to the response of the reviews)

p. 6, l. 10: 'calculated to mass absorbance' – is there a word missing (e.g. 'obtain')?

p. 9, l. 3: What is a 'carry over for sulfate'

p. 11, l. 11: 'numeric number' is redundant

Figure 2, y-axis should be 'Uptake coefficient'

---

## Author Response (AR2)

**Response to Editor's comments (Manuscript Ref. NO.: acp-2017-120)**

We appreciate the editor for the thoughtful comments and guidance. The manuscript has been carefully checking for the errors and the consistency in model description. The responses to the comment are shown below.

**1) Referee #1, Comment 1:** The partitioning between the gas and condensed phase are treated in a similar way, despite being fundamentally different in nature. For a solid surface, the adsorption and desorption processes do follow a different formalism, typically through a Langmuir-Hinshelwood formalism which takes into account a given number of adsorption sites. The consequences is that adsorption decreases with time or increasing concentration, while here it is simulated in a constant way with time. how can you justify such an assumption? Also products such as sulfate are probably staying on the surface, thereby also using adsorption sites i.e., poisoning the surface. How would your model change if you implement such time/concentration dependence?

**Previous Response:** We assumed that the gas-particle partitioning onto dust is operated by an absorption process (Eq. 7) by several reasons (see section 3.2.1). First, unlike pure metal oxide which is governed by the adsorptive partitioning, the composition of authentic mineral dust such as Arizona Test Dust (ATD) is complex. The fresh ATD contains inorganic salts that are hygroscopic and form the water film above efflorescence relative humidity or deliquescence relative humidity. Second, the partitioning process is dynamic due to the formation of various hygroscopic salts of sulfate and nitrate due to the reaction of alkaline carbonates and metal oxides with inorganic acids (sulfuric acid and nitric acid). Third, the sulfate formation in our study increased as increasing humidity due to the dissolution of tracers into the water layer (see section 3.2.1). If partitioning is processed by the adsorptive mode, water molecules compete for the site with tracers and reduce partitioning of tracers (Cwiertny et al., 2008). The amount of the surface water on dust particles, which was measured using FTIR (submitted in the other journal), was multi-layered.

**Editor:** It seems to me that the only change in response to this comment was that you replaced all 'adsorption' by 'absorption' in the manuscript. However, this raises more questions than it adds to clarification. You argue that the hygroscopic fraction of the dust forms an aqueous phase where then the chemical processing in the 'dust phase' occurs. Thus, the extent of this processing will depend on the mass (volume) of the hygroscopic dust material. It is not clear to me why parameters such as the absorption rate constants (e.g. R5 and Equation 5) can be used that relate to a particle surface (m-2) and not to a (partial) particle volume. This should be clarified throughout the manuscript. The text should be carefully checked for consistency (e.g. p. 15, l. 10 'adsorption-desorption') and conclusions should be refined (e.g. why is BET surface area needed (p. 20, l. 26) if the hygroscopic material determines the reactions medium?)

**Response to Editor:** We agree with the Editor's view. In the revised manuscript, we use word "adsorption-desorption" for gas-dust partitioning of tracers (i.e., $SO_2$ or $NO_2$). Unlike partitioning on pure metal oxide, gas-dust partitioning is processed on the multilayer coated dust with water molecules. In order to clarify this, the sentence was added into the revised manuscript and reads now, "The partitioning processes between the gas phase and multilayer coted dust were treated by the adsorption–desorption kinetic mechanism." (1[st] paragraph in Section 3). Please also see the 1[st] paragraph of Section 3.2.1 Gas–dust particle partitioning.

**2) Referee #1, Comment 4:** Too many rate constants are estimated without any justification. Please justify and explain your estimations.

**Response:** Most of the rate constants shown in Table 3 were estimated using the indoor chamber data obtained in the previous study (Park and Jang, 2016). The rate constants of R10 (electron-hole production) and R11 (recombination of electron-hole) in the manuscript is estimated using Eq. 10 (photoactivation rate, JATD) in the manuscript (Section 3.2.3). The rate constant of R13 (reaction of $SO_2$ with dust-phase OH radicals) is set to the same reaction rate constant for the reaction of $SO_2$ with OH radicals in gas phase. Without sunlight, autoxidation of $SO_2$ (R9) is dominant in dust phase and its rate constant was obtained from indoor chamber data under various humidity conditions (Exp. D1-D3 in Table 1). With sunlight, the photochemical reaction is the major source for sulfate production. Using the same approach with autoxidation, the rate constant of R12 was estimated under different humidity conditions. Also, the rate constants of R14 (heterogeneous autoxidation of $SO_2$ in the presence of ozone) and R15 (heterogeneous oxidation of $O_3$) were estimated using experiments D4 and L5 in Table 1, respectively. The rate constants of R18 (heterogeneous autoxidation of $NO_2$) and R19 (heterogeneous photocatalytic oxidation of $NO_2$) were estimated using experiments D5 and L7 in Table 1, respectively.

**Editor:** In my opinion, your response is neither a justification nor an explanation why so many rate constants were estimated. (a) Please add a more detailed discussion of uncertainties and background on these constants in order to fulfill the reviewer's inquiry. (b) Also, I noticed that e.g. $K_{d,SO2}$ is used for dry particles – how can this be justified?

**Response to Editor:**
(a) Thank you much for the editor's comment on uncertainties of model parameter and the prediction of sulfate. The third paragraph in Section 5 and Figure S7 were revised to illustrate the uncertainty in major model parameters ([H[+]], $F_{water}$, $K_{d, SO_2}$, $k_{auto}$ and $k_{OH,O_2}$) and the formation of sulfate ([$SO_4^{2-}$]$_T$). Please also notice the figure caption in Figure S7 for the detailed explanation.

The rate constants of heterogeneous photooxidation reactions of $SO_2$ and $NO_2$ on airborne mineral dust were unknown. In order to obtain the rate constants for the dust phase chemistry, we ran our model using the indoor data reported by Park and Jang (2016). In the 2[nd] paragraph of Section "3

AMAR model description", we summarized the source data used for processing major model parameters.

"The rate constants associated with various reaction mechanisms in the AMAR model were determined by simulating pre-existing indoor chamber data obtained from controlled experimental conditions (Park and Jang, 2016). For example, the rate constant ($k_{auto}$, s$^{-1}$) for SO$_2$ autoxidation is semiempirically determined by fitting the predicted concentration of sulfate to the experimental data D1 in Table 1. The gas–dust partitioning constant ($K_{d,SO_2}$, Sect. 3.2.1) of SO$_2$ is dependent of temperature, aerosol water content, and acidity. $K_{d,SO_2}$ values were semiempirically determined using data D1-D3 (three different RHs) and the literature parameters related to the effect of temperature and acidity on $K_{d,SO_2}$. The rate constant ($k_{photo}$, cm$^3$ molecule$^{-1}$ s$^{-1}$) for sulfate formation by photocatalytic reactions is semiempirically determined using data L1-L3 (three different RHs) in Table 1. In the presence of ozone, $k_{auto}$ and $k_{photo}$ are determined using data D4 and L4, respectively."

(b) The value of $K_{d,SO_2}$ at 20% RH was obtained from literature data (Adams et al. 2005; Huang et al. 2015). In order to estimate $K_{d,SO_2}$ at different RH, data D1-D3 in Table 1 (three different RHs) were used to estimate the effects of humidity on the gas-dust partitioning process. Thus, $K_{d,SO_2}$ is a function of RH.

**3) Referee #2, Comment 2:** In addition to react with SO$_2$ and NO$_2$, OH radicals produced on the surface of particles under UV conditions can undergo heterogeneous reaction with particles as well as self-reactions, resulting in the significant decrease of OH radicals participate in the oxidation of SO$_2$ and NO$_2$, and subsequently overestimating sulfate and nitrate formation in the model. Furthermore, in addition to compete OH radicals with SO$_2$, the presence of NO$_2$ can also react with SO$_2$ on the surface of particles to promote sulfate formation at high RHs as like in aqueous phase. However, these mechanisms were not considered in dust phase in the model (Table S1).

**Response:** In our model, the apparent rate constant of the formation of the dust-phase OH radicals is estimated using indoor chamber data. The synergistic effect of NO$_2$ on sulfate formation under UV light is explained by the HONO production through the reaction of NO$_2$ with electrons or holes in dust phase (R16). HONO will then be decomposed via photolysis to form OH radicals (R17).

**Editor:** The reviewer's concern of missing reactions in your mechanism is well justified. How would the recombination of OH or their loss on particle surfaces affect your results? Please add a discussion about what it is known about such processes and to what extent they may compete with other processes in your system.

**Response to Editor:** In current our knowledge, the amount and the type of conductive metal oxides in ATD particles are unclear. In our model, the formation of OH radicals on dust is operated using apparent rate constants for the formation and decay of electron-hole (R10 and R11). The mechanistic role of the catalytic formation of the electron-hole pair (R10) and their

recombination (R11) can compensate the formation and the self-reaction of OH radicals (Section 3.2.3).

**4) Difference between dust chemistry and aqueous phase chemistry**
**Editor:** Related to the comment above, I do not understand the fundamental difference between the aqueous phase chemistry (Section 3.1.3) and dust chemistry (Section 3.2). Both are reactions that occur in a bulk aqueous phase and thus mechanistically they should be treated equally (even though different chemical reactions occur). Please justify the differentiation into two types of processes. It should be stated throughout the manuscript that the 'dust phase' is also technically an aqueous phase

**Response to Editor:** Please also find the response to editor's comment 1. The aqueous phase reaction is processed in a bulk phase while dust chemistry occurs in the multilayer comprising electrolytes and water on dust surfaces. Hence, the estimation of the water content on dust surfaces, which is influenced by hygroscopic properties of dust surfaces, temperature, and acidity, is essential as discussed in Section 3.2.1. The water content on dust surfaces is also dependent of the amount of dust (relevant to the surface area) and dust compositions.

**5) Use of Henry's law constants**
**a)** What is the ionic strength and/or acidity of the aqueous phases? Is the application of Henry's law constants (for ideal solutions) justified? If not, how does this affect the results and the possibility of extrapolation of the derived rate constants to other conditions?

**Response to Editor:** In AMAR, aerosol acidity ($[H^+]$, mol $L^{-1}$) is estimated at each time step by E-AIM II (Clegg et al., 1998;Wexler and Clegg, 2002;Clegg and Wexler, 2011) corrected for the ammonia rich condition (Li and Jang, 2012; Li et al., 2015;Beardsley and Jang, 2016) as a function of inorganic composition measured by PILS-IC (Section 3.1.3).

For the highly concentrated electrolyte aerosol, the deviation of the compound's solubility predicted using Henry's constants from the actual solubility would be varied depending upon the chemical structure. In the current knowledge, we do not know the actual Henry's constant of each species in the highly concentrated electrolyte solution. For aqueous phase reactions (no dust), Henry's constants that are reported in the modeling paper by Liang and Jocobson (1999) were applied to the AMAR model. Although this implementation can be potentially problematic to predict sulfate production, our model simulation reasonably predicted outdoor chamber data (Figure 3(a) in Section 4.1)

For dust heterogeneous chemistry, Henry's constants of various tracers were used to scale their gas-dust partitioning coefficient based on the known value for $SO_2$ on ATD dust particles (Section 3.2.1). For example, the literature value for the gas-ATD partitioning coefficient of SO2 was 1.3 $m^3$ $m^{-2}$ at 20% RH. The gas-ATD partitioning coefficient of $SO_2$ is much greater than Henry's constant by several orders ($10^5$) when the same unit is applied to both constants ($m^3/\mu g$). The scaling of the gas-ATD partitioning coefficients of the tracers of this study using Henry's

constants may cause some inaccuracy in the estimation of the concentration of adsorbed tracers due to the difference in the activity coefficient of each compound in different media (dust surface vs. dilute aqueous phase). As shown in Figure 3 (Section 4.1), the sulfate production in the presence of ATD particles reasonably accorded with chamber data.

**b)** Is the pH sufficiently low that the uptake of $SO_2$ can be indeed solely described by the physical Henry's law constant $K_{H,SO2}$? This can be only applied if the solution is sufficiently acidic and no dissociation occurs; otherwise the effective Henry's law constant including dissociation should be included. Please justify.

**Response to Editor:** As shown in Eq. (7), the impact of aerosol acidity on the gas-dust partitioning of $SO_2$ is treated in the model. When pH is low, the uptake of $SO_2$ is still influenced by humidity and aerosol compositions for both aqueous phase reactions and dust heterogeneous chemistry. As shown in Figure 5, the humidity conditions can significantly influence sulfate formation due to the higher uptake of $SO_2$ at the higher RH. In general, aerosol acidity has a compounding effect because the higher acidity can reduce the solubility of $SO_2$ (Eq. (7)) but increase hygroscopic property of aerosol. For aqueous phase reactions, we included the uptake process and acid dissociation reactions for $HO_2$, $HCOOH$, $HONO$ and $HCHO$ (acid dissociation reactions in Table S1). In addition to reactions of inorganic species, the influence of organic species (i.e., $HCOOH$, $HCHO$, and $CH_3CHO$) on dust heterogeneous chemistry needs to be investigated in the future (last paragraph in Section 6).

**6) Language**
The whole manuscript should be carefully checked for proper use of English language. I list some rather unusual or unclear expressions below (line numbers refer to the marked-up manuscript that was attached to the response of the reviews)
p. 6, l. 10: 'calculated to mass absorbance' – is there a word missing (e.g. 'obtain')?
p. 9, l. 3: What is a 'carry over for sulfate'
p. 11, l. 11: 'numeric number' is redundant
Figure 2, y-axis should be 'Uptake coefficient'

**Response to Editor:** The errors were corrected. The manuscript has been thoroughly checked for grammars and spelling.

**Marked–up manuscript**

[revised manuscript text omitted]

---

## Editor Decision (ED2)

Response to Editor's comments (Manuscript Ref. NO.: acp-2017-120)

We appreciate the editor for the thoughtful comments and guidance. The manuscript has been carefully checking for the errors and the consistency in model description. The responses to the comment are shown below.

1) Referee #1, Comment 1: The partitioning between the gas and condensed phase are treated in a similar way, despite being fundamentally different in nature. For a solid surface, the adsorption and desorption processes do follow a different formalism, typically through a Langmuir-Hinshelwood formalism which takes into account a given number of adsorption sites. The consequences is that adsorption decreases with time or increasing concentration, while here it is simulated in a constant way with time. how can you justify such an assumption? Also products such as sulfate are probably staying on the surface, thereby also using adsorption sites i.e., poisoning the surface. How would your model change if you implement such time/concentration dependence?

Previous Response: We assumed that the gas-particle partitioning onto dust is operated by an absorption process (Eq. 7) by several reasons (see section 3.2.1). First, unlike pure metal oxide which is governed by the adsorptive partitioning, the composition of authentic mineral dust such as Arizona Test Dust (ATD) is complex. The fresh ATD contains inorganic salts that are hygroscopic and form the water film above efflorescence relative humidity or deliquescence relative humidity. Second, the partitioning process is dynamic due to the formation of various hygroscopic salts of sulfate and nitrate due to the reaction of alkaline carbonates and metal oxides with inorganic acids (sulfuric acid and nitric acid). Third, the sulfate formation in our study increased as increasing humidity due to the dissolution of tracers into the water layer (see section 3.2.1). If partitioning is processed by the adsorptive mode, water molecules compete for the site with tracers and reduce partitioning of tracers (Cwiertny et al., 2008). The amount of the surface water on dust particles, which was measured using FTIR (submitted in the other journal), was multi-layered.

Editor: It seems to me that the only change in response to this comment was that you replaced all 'adsorption' by 'absorption' in the manuscript. However, this raises more questions than it adds to clarification. You argue that the hygroscopic fraction of the dust forms an aqueous phase where then the chemical processing in the 'dust phase' occurs. Thus, the extent of this processing will depend on the mass (volume) of the hygroscopic dust material. It is not clear to me why parameters such as the absorption rate constants (e.g. R5 and Equation 5) can be used that relate to a particle surface (m-2) and not to a (partial) particle volume. This should be clarified throughout the manuscript. The text should be carefully checked for consistency (e.g. p. 15, l. 10 'adsorption-desorption') and conclusions should be refined (e.g. why is BET surface area needed (p. 20, l. 26) if the hygroscopic material determines the reactions medium?)

Response to Editor: We agree with the Editor's view. In the revised manuscript, we use word "adsorption-desorption" for gas-dust partitioning of tracers (i.e., SO2 or NO2). Unlike partitioning on pure metal oxide, gas-dust partitioning is processed on the multilayer coated dust with water molecules. In order to clarify this, the sentence was added into the revised manuscript

and reads now, "The partitioning processes between the gas phase and multilayer coted dust were treated by the adsorption–desorption kinetic mechanism." (1st paragraph in Section 3). Please also see the 1st paragraph of Section 3.2.1 Gas–dust particle partitioning.

Editor, 2$^{nd}$ round of comments: I am more confused than before now. Initially, the reviewer had pointed out the fundamental differences between adsorption and absorption with different formalisms. The reviewer explained that

*"The consequences is that adsorption decreases with time or increasing concentration, while here it is simulated in a constant way with time. how can you justify such an assumption? Also products such as sulfate are probably staying on the surface, thereby also using adsorption sites i.e., poisoning the surface."*

Instead of addressing this point and explaining possible consequences of surface effects (such as poisoning) you simply changed 'adsorption' to 'absorption' throughout the manuscript and argued that the description of an aqueous phase process is justified. Now, in the newly revised version, everything was changed back to 'adsorption' but the reviewer's comment was not addressed at all. Please, justify clearly the formalism you use and use a constant description of the processes so that the equations that describe your experimental system make sense and can be applied by other researchers.

Throughout the revised manuscript, you are using both 'abs' (e.g., p. 7) and 'ads' (e.g., p. 10) as indices for constants.

Please make sure that your discussion is consistent and address the reviewer's comments.

4) Difference between dust chemistry and aqueous phase chemistry

Editor: Related to the comment above, I do not understand the fundamental difference between the aqueous phase chemistry (Section 3.1.3) and dust chemistry (Section 3.2). Both are reactions that occur in a bulk aqueous phase and thus mechanistically they should be treated equally (even though different chemical reactions occur). Please justify the differentiation into two types of processes. It should be stated throughout the manuscript that the 'dust phase' is also technically an aqueous phase

Response to Editor: Please also find the response to editor's comment 1. The aqueous phase reaction is processed in a bulk phase while dust chemistry occurs in the multilayer comprising electrolytes and water on dust surfaces. Hence, the estimation of the water content on dust surfaces, which is influenced by hygroscopic properties of dust surfaces, temperature, and acidity, is essential as discussed in Section 3.2.1. The water content on dust surfaces is also dependent of the amount of dust (relevant to the surface area) and dust compositions.

Editor, 2$^{nd}$ round of comments: Here you argue again that the amount of water is dependent on the amount of dust – which is contradictory to the assumption of adsorption processes but rather points to an absorption process.

Please clarify and discuss consistently throughout the manuscript what kind of processes you are considering, what type of parameters are used and how they can be extrapolated and applied to other conditions so that your claim in the abstract

*"The AMAR model, derived in this study with ATD particles, will provide a platform for predicting sulfate formation in the presence of authentic dust particles (e.g. Gobi and 30 Saharan dust)."*

is justified.

5) Use of Henry's law constants

a) What is the ionic strength and/or acidity of the aqueous phases? Is the application of Henry's law constants (for ideal solutions) justified? If not, how does this affect the results and the possibility of extrapolation of the derived rate constants to other conditions?

Response to Editor: In AMAR, aerosol acidity ($[H+]$, mol L-1) is estimated at each time step by E-AIM II (Clegg et al., 1998;Wexler and Clegg, 2002;Clegg and Wexler, 2011) corrected for the ammonia rich condition (Li and Jang, 2012; Li et al., 2015;Beardsley and Jang, 2016) as a function of inorganic composition measured by PILS-IC (Section 3.1.3).

For the highly concentrated electrolyte aerosol, the deviation of the compound's solubility predicted using Henry's constants from the actual solubility would be varied depending upon the chemical structure. In the current knowledge, we do not know the actual Henry's constant of each species in the highly concentrated electrolyte solution. For aqueous phase reactions (no dust), Henry's constants that are reported in the modeling paper by Liang and Jocobson (1999) were applied to the AMAR model. Although this implementation can be potentially problematic to predict sulfate production, our model simulation reasonably predicted outdoor chamber data (Figure 3(a) in Section 4.1)

For dust heterogeneous chemistry, Henry's constants of various tracers were used to scale their gas-dust partitioning coefficient based on the known value for SO2 on ATD dust particles (Section 3.2.1). For example, the literature value for the gas-ATD partitioning coefficient of SO2 was 1.3 m3 m-2 at 20% RH. The gas-ATD partitioning coefficient of SO2 is much greater than Henry's constant by several orders (105) when the same unit is applied to both constants (m3/μg). The scaling of the gas-ATD partitioning coefficients of the tracers of this study using Henry's constants may cause some inaccuracy in the estimation of the concentration of adsorbed tracers due to the difference in the activity coefficient of each compound in different media (dust surface vs. dilute aqueous phase). As shown in Figure 3 (Section 4.1), the sulfate production in the presence of ATD particles reasonably accorded with chamber data.

Editor, 2[nd] round of comments: The fact that your model can predict sulfate production reasonably well is not a proof that assuming that the Henry's law constant is correct. There could be cancelling effects that lead to the correct result for the wrong reason. Given the fact that you estimated several rate constants, the whole set of constants (partitioning, rate constants, Henry's

law constants) might reproduce the observations; however, they might not properly represent the individual processes. If someone will use your model and apply it to other conditions, this will lead to biases in predicted sulfate formation.

---

## Author Response (AR3)

Response to Editor's comments (Manuscript Ref. NO.: acp-2017-120)

We thank the editor for the considerate comments. We have carefully revised the manuscript. The responses to the Editor's 2nd round comments are shown as below.

**Comment 1**

**Originally from comment 1 of Referee #1:** The partitioning between the gas and condensed phase are treated in a similar way, despite being fundamentally different in nature. For a solid surface, the adsorption and desorption processes do follow a different formalism, typically through a Langmuir-Hinshelwood formalism which takes into account a given number of adsorption sites.

The consequences is that adsorption decreases with time or increasing concentration, while here it is simulated in a constant way with time. how can you justify such an assumption? Also products such as sulfate are probably staying on the surface, thereby also using adsorption sites i.e., poisoning the surface. How would your model change if you implement such time/concentration dependence?

**Our Response**: We assumed that the gas-particle partitioning onto dust is operated by an absorption process (Eq. 7) by several reasons (see section 3.2.1). First, unlike pure metal oxide which is governed by the adsorptive partitioning, the composition of authentic mineral dust such as Arizona Test Dust (ATD) is complex. The fresh ATD contains inorganic salts that are hygroscopic and form the water film above efflorescence relative humidity or deliquescence relative humidity. Second, the partitioning process is dynamic due to the formation of various hygroscopic salts of sulfate and nitrate due to the reaction of alkaline carbonates and metal oxides with inorganic acids (sulfuric acid and nitric acid). Third, the sulfate formation in our study increased as increasing humidity due to the dissolution of tracers into the water layer (see section 3.2.1). If partitioning is processed by the adsorptive mode, water molecules compete for the site with tracers and reduce partitioning of tracers (Cwiertny et al., 2008). The amount of the surface water on dust particles, which was measured using FTIR (submitted in the other journal), was multi-layered.

**Editor's comment to our response**: It seems to me that the only change in response to this comment was that you replaced all 'adsorption' by 'absorption' in the manuscript. However, this raises more questions than it adds to clarification. You argue that the hygroscopic fraction of the dust forms an aqueous phase where then the chemical processing in the 'dust phase' occurs. Thus, the extent of this processing will depend on the mass (volume) of the hygroscopic dust material. It is not clear to me why parameters such as the absorption rate constants (e.g. R5 and Equation 5) can be used that relate to a particle surface (m-2) and not to a (partial) particle volume. This should be clarified throughout the manuscript. The text should be carefully checked for consistency (e.g. p. 15, l. 10 'adsorption-desorption') and conclusions should be refined (e.g. why is BET surface area needed (p. 20, l. 26) if the hygroscopic material determines the reactions medium?)

**Response to Editor's comment**: We agree with the Editor's view. In the revised manuscript, we use word "adsorption-desorption" for gas-dust partitioning of tracers (i.e., $SO_2$ or $NO_2$). Unlike partitioning on pure metal oxide, gas-dust partitioning is processed on the multilayer coated dust

with water molecules. In order to clarify this, the sentence was added into the revised manuscript and reads now, "The partitioning processes between the gas phase and multilayer coted dust were treated by the adsorption–desorption kinetic mechanism." (1st paragraph in Section 3). Please also see the 1st paragraph of Section 3.2.1 Gas–dust particle partitioning.

**Editor, 2nd round of comments**: I am more confused than before now. Initially, the reviewer had pointed out the fundamental differences between adsorption and absorption with different formalisms. The reviewer explained that

"*The consequences is that adsorption decreases with time or increasing concentration, while here it is simulated in a constant way with time. how can you justify such an assumption? Also products such as sulfate are probably staying on the surface, thereby also using adsorption sites i.e., poisoning the surface.*"

Instead of addressing this point and explaining possible consequences of surface effects (such as poisoning) you simply changed 'adsorption' to 'absorption' throughout the manuscript and argued that the description of an aqueous phase process is justified. Now, in the newly revised version, everything was changed back to 'adsorption' but the reviewer's comment was not addressed at all. Please, justify clearly the formalism you use and use a constant description of the processes so that the equations that describe your experimental system make sense and can be applied by other researchers.

Throughout the revised manuscript, you are using both 'abs' (e.g., p. 7) and 'ads' (e.g., p. 10) as indices for constants.

Please make sure that your discussion is consistent and address the reviewer's comments.

**Response to the 2nd round comment from Editor:** Thank Editor for the thoughtful comment on the justification of the uptake process of gaseous compounds on dust particles. We have carefully thought about the partitioning process in our model. Unlike pure metal oxide particle, authentic dust particles are complex mixtures comprising metal oxides, alkaline carbonates, and alkaline sulfate. Sulfate salts are water soluble and phase transition as a function of humidity. Some salts such as sulfate of magnesium and calcium can be hydrated even at low humidity (Beardsley et al., 2013; Jang et al., 2010). Gustafsson et al. (2005) reported that ATD particles showed a substantially high affinity to water compared to pure $CaCO_3$ particles. In their study, the water content of ATD particles, which is measured using the thermogravimetric analysis (TGA), ranges from two monolayers to four monolayers based on the BET surface area between 20%-80% relative humidity. In particular, hydrophilic tracer compounds such as $SO_2$ and HONO are sensitive to the water content of dust particles. The amount of $SO_2$ on dust particles increases with increasing humidity due to the partitioning of $SO_2$ to water layers. Therefore, we assumed that gas-particle partitioning onto dust (e.g., ATD) is processed in absorption mode. Besides, inorganic acids can be dissociated in the multilayer water on dust and affect partitioning of $SO_2$ (Section 3.2.1). The first paragraph in Section 3 (AMAR model description) was modified in the manuscript and reads now: "*ATD particles are known to be coated with the multilayer of water*

*due to their high affinity to water (Gustafsson et al., 2005) (Sect. 3.2.1). Therefore, we assumed that gas–dust partitioning of tracers on multilayer water is processed in absorption mode.*" Also sentence "*Some salts such as sulfate of magnesium and calcium can be hydrated even at low humidity (Beardsley et al., 2013;Jang et al., 2010). Gustafsson et al. (2005) reported that ATD particles showed a substantially high affinity to water compared to pure $CaCO_3$ particles. In their study, the water content of ATD particles, which is measured using the thermogravimetric method, ranges from two monolayers to four monolayers based on the BET surface area between 20%– 80% relative humidity.*" was added to the first paragraph of section 3.2.1.

The BET surface area is generally important for Langmuir-Hinshelwood formalism which takes into account a certain number of adsorption sites within a monolayer. In the presence of multilayer water on dust, the adsorption mode, which is parameterized by the BET surface area, is inappropriate to process dust chemistry. Multilayer water can fill the void volume on the dust surface decreasing tortuosity. Thus, we think that the geometric surface area is more proper to describe dust chemistry and partitioning in the presence of multilayer water. For example, the partitioning process of this study is normalized by the geometric surface area (Eq. 5) instead of the BET area. All absorption rates for tracers (e.g., Eq. R7) are also processed using the geometric surface area. The second and third term in Eq. 8, which estimates the water content in dust particles, were also normalized using the geometric surface area ($f_{dust,mass\_to\_surface}$ x $A_{dust}$ = dust mass concentration). Sentence "*To extend the AMAR model to other dust materials, the molecular level surface area (BET surface area) should be considered in the future.*" has been deleted in the revised manuscript.

**Comment 2: Difference between dust chemistry and aqueous phase chemistry**

**The 1st round comment from Editor:** Related to the comment above, I do not understand the fundamental difference between the aqueous phase chemistry (Section 3.1.3) and dust chemistry (Section 3.2). Both are reactions that occur in a bulk aqueous phase and thus mechanistically they should be treated equally (even though different chemical reactions occur). Please justify the differentiation into two types of processes. It should be stated throughout the manuscript that the 'dust phase' is also technically an aqueous phase

**Response to the 1st round comment from Editor**: Please also find the response to comment 1 above. The aqueous phase reaction is processed in a bulk phase while dust chemistry occurs in the multilayer comprising electrolytes and water on dust surfaces. Hence, the estimation of the water content on dust surfaces, which is influenced by hygroscopic properties of dust surfaces, temperature, and acidity, is essential as discussed in Section 3.2.1. The water content on dust surfaces is also dependent of the amount of dust (relevant to the surface area) and dust compositions.

**Editor, 2nd round of comments**: Here you argue again that the amount of water is dependent on the amount of dust – which is contradictory to the assumption of adsorption processes but rather points to an absorption process.

Please clarify and discuss consistently throughout the manuscript what kind of processes you are considering, what type of parameters are used and how they can be extrapolated and applied to other conditions so that your claim in the abstract

*"The AMAR model, derived in this study with ATD particles, will provide a platform for predicting sulfate formation in the presence of authentic dust particles (e.g. Gobi and 30 Saharan dust)."* is justified.

**Response to the 2ⁿᵈ round comment from Editor**: Please also find the response to the comment 1 above. In this model, dust phase kinetics is approached by absorption mode.

In the current stage, it is uncertain how quickly and how deeply water molecules can penetrate inside dust particles. As the editor commented in the 1ˢᵗ round review, dust chemistry within multi-layer of water can be treated similarly with aqueous chemistry. However, the aqueous chemistry is processed through the whole aerosol volume and the dust chemistry is processed in the water layers on the surface of dust particles. The dust absorption model is dependent on the volume of multilayer water which is related to the geometric surface area of particles. For the computation using the chemical solver, the partitioning of tracers between the gas phase and dust particles is approached by on- (collision onto the particle surface) and off- (desorption) modes. The desorption rate is dependent on water content (Section 3.2.1) suggesting that partitioning is governed by the absorption mode. In this revision, the symbol "*abs*" is used for dust phase chemistry instead of "*ads*". Sentence "*Overall, dust chemistry within multilayer of water is treated by the similar manner to aqueous chemistry. However, the aqueous chemistry is operated through the whole aerosol volume and the dust chemistry is processed in the water layers on the surface of dust particles.*" was added to the first paragraph of section 3 (AMAR model description).

The last sentence in the abstract has been removed. The application of the current model of this study into other dust particle systems has been explained in the last paragraph of Section 6 (Conclusion and Atmospheric Implication) and reads now, "*To extend the AMAR model derived with ATD particle to the prediction of sulfate in the presence of ambient dust particles, the model parameters related to rate constants and the physical characteristics (e.g. surface area and hygroscopic properties) of dust particles need to be modulated with laboratory experiments. The photoactivation rate constant ( $k_{e\_h}^{j}$ in Section 3.2.3) to form electron-hole pairs should be revisited to apply the model to different mineral dust systems, which are different from ATD in the photocatalytic capacity of conductive metal oxides. In addition to reactions of inorganic species, the influence of organic species (i.e., HCOOH, HCHO, and CH₃CHO) on dust heterogeneous chemistry needs to be investigated in the future.*"

**Comment 3: Use of Henry's law constants**

**1ˢᵗ Round comment form Editor:** a) What is the ionic strength and/or acidity of the aqueous phases? Is the application of Henry's law constants (for ideal solutions) justified? If not, how does this affect the results and the possibility of extrapolation of the derived rate constants to other conditions?

**Response to the 1st round comment from Editor:** In AMAR, aerosol acidity ([H+], mol L-1) is estimated at each time step by E-AIM II (Clegg et al., 1998;Wexler and Clegg, 2002;Clegg and Wexler, 2011) corrected for the ammonia rich condition (Li and Jang, 2012; Li et al., 2015;Beardsley and Jang, 2016) as a function of inorganic composition measured by PILS-IC (Section 3.1.3).

For the highly concentrated electrolyte aerosol, the deviation of the compound's solubility predicted using Henry's constants from the actual solubility would be varied depending upon the chemical structure. In the current knowledge, we do not know the actual Henry's constant of each species in the highly concentrated electrolyte solution. For aqueous phase reactions (no dust), Henry's constants that are reported in the modeling paper by Liang and Jocobson (1999) were applied to the AMAR model. Although this implementation can be potentially problematic to predict sulfate production, our model simulation reasonably predicted outdoor chamber data (Figure 3(a) in Section 4.1)

For dust heterogeneous chemistry, Henry's constants of various tracers were used to scale their gas-dust partitioning coefficient based on the known value for $SO_2$ on ATD dust particles (Section 3.2.1). For example, the literature value for the gas-ATD partitioning coefficient of $SO_2$ was 1.3 $m^3$ $m^{-2}$ at 20% RH. The gas-ATD partitioning coefficient of $SO_2$ is much greater than Henry's constant by several orders (105) when the same unit is applied to both constants (m3/µg). The scaling of the gas-ATD partitioning coefficients of the tracers of this study using Henry's constants may cause some inaccuracy in the estimation of the concentration of adsorbed tracers due to the difference in the activity coefficient of each compound in different media (dust surface vs. dilute aqueous phase). As shown in Figure 3 (Section 4.1), the sulfate production in the presence of ATD particles reasonably accorded with chamber data.

**Editor, 2nd round of comments**: The fact that your model can predict sulfate production reasonably well is not a proof that assuming that the Henry's law constant is correct. There could be cancelling effects that lead to the correct result for the wrong reason. Given the fact that you estimated several rate constants, the whole set of constants (partitioning, rate constants, Henry's law constants) might reproduce the observations; however, they might not properly represent the individual processes. If someone will use your model and apply it to other conditions, this will lead to biases in predicted sulfate formation.

**Response to the 2nd round comment from Editor**: We agree that the Henry's law constant may lead to biases for uptake processes on the dust particle. The partitioning constant of tracers should be clarified in the future. To respond to the comments, we add the new sentence into the last paragraph of section 5 (Sensitivity and uncertainties): "*
[revised manuscript text omitted]